# Deciphering the allosteric regulation of mycobacterial inosine-5′-monophosphate dehydrogenase

Ondřej Bulvas [1], Zdeněk Knejzlík[1], Jakub Sýs [1], Anatolij Filimoněnko[1], Monika Čížková [1], Kamila Clarová[1], Dominik Rejman [1], Tomáš Kouba [1] ✉ & Iva Pichová [1] ✉

Allosteric regulation of inosine 5′-monophosphate dehydrogenase (IMPDH), an essential enzyme of purine metabolism, contributes to the homeostasis of adenine and guanine nucleotides. However, the precise molecular mechanism of IMPDH regulation in bacteria remains unclear. Using biochemical and cryo-EM approaches, we reveal the intricate molecular mechanism of the IMPDH allosteric regulation in mycobacteria. The enzyme is inhibited by both GTP and (p)ppGpp, which bind to the regulatory CBS domains and, via interactions with basic residues in hinge regions, lock the catalytic core domains in a compressed conformation. This results in occlusion of inosine monophosphate (IMP) substrate binding to the active site and, ultimately, inhibition of the enzyme. The GTP and (p)ppGpp allosteric effectors bind to their dedicated sites but stabilize the compressed octamer by a common mechanism. Inhibition is relieved by the competitive displacement of GTP or (p)ppGpp by ATP allowing IMP-induced enzyme expansion. The structural knowledge and mechanistic understanding presented here open up new possibilities for the development of allosteric inhibitors with antibacterial potential.

Purine metabolism provides the fundamental building blocks and energy required for biological processes. Central to the purine metabolic pathway is inosine-5′-monophosphate dehydrogenase (IMPDH; EC 1.1.1.205), a rate-limiting enzyme catalyzing the NAD$^+$-dependent oxidation of inosine-5′-monophosphate (IMP) to xanthosine 5′-monophosphate (XMP)[1]. IMPDH, located at the crossroads of ATP and GTP biosynthetic branches, has been studied as a natural regulation point. Despite extensive investigations into the allosteric regulation of IMPDH in various organisms, including eukaryotes and certain bacteria[2,3], our understanding of the molecular details governing the regulation of mycobacterial IMPDH remains limited. Mycobacterial IMPDH is considered by some authors to be a promising target for the treatment of infections caused by pathogenic *Mycobacterium* species[4–6], although others have raised concerns about the efficacy of such a potential treatment[7].

Most of the IMPDHs share a conserved two-domain architecture consisting of a catalytic domain and a smaller regulatory cystathionine β-synthase (CBS) domain (also known as the Bateman module). The relative orientation between the catalytic and CBS domains varies in different IMPDH structures due to the presence of hinge regions formed by highly flexible linkers. In solution, IMPDH can exist as tetramers or octamers[8], with catalytic domains located at the core of the assembly. While IMPDH tetramers are exclusively formed through the interface of catalytic domains, the octamers are further stabilized through CBS-CBS domain interactions at the periphery.

The CBS domain serves as an allosteric regulatory module responding to the binding of nucleotide effectors[9]. Alexandre et al.[8] propose a classification of bacterial IMPDHs, dividing them into two classes. Class I IMPDHs require ATP to bind to the CBS domain for full activity and always exist in an octameric form. In contrast, class II

[1]Institute of Organic Chemistry and Biochemistry of the Czech Academy of Sciences, Prague, Czech Republic. ✉e-mail: tomas.kouba@uochb.cas.cz; iva.pichova@uochb.cas.cz

enzymes, which occur as tetramers, do not require ligand binding for activation and undergo ATP-induced octamerization. In general, nucleotides bind to the CBS domain at two conserved canonical binding sites and, in some cases, at additional non-canonical sites[10]. Canonical Site 1 of bacterial IMPDHs exclusively binds ATP, while Site 2 can competitively bind ATP, GTP or guanosine diphosphate (GDP)[11]. The enzyme activity of bacterial IMPDHs is inhibited only when ATP is bound to Site 1 and guanine nucleotide is bound to Site 2. This unusual requirement for a simultaneous combination of two different effectors for the allosteric regulation of bacterial IMPDHs has only recently been recognized. In addition, most bacterial phyla appear to possess an additional non-canonical (p)ppGpp binding pocket that partially overlaps with the canonical Site 2[11].

Recently, it has been suggested that bacterial IMPDHs have diversified into two groups: one exclusively using (p)ppGpp, and the other relying solely on GTP or GDP as a negative regulator[2]. According to this classification, mycobacterial IMPDH should only be regulated by a combination of ATP and (p)ppGpp. However, canonical Site 2, which is important for GTP and GDP binding, seems to be conserved in most bacteria, including mycobacteria (Supplementary Fig. 1). This suggests that GTP or GDP might also play a role in affecting the activity of mycobacterial IMPDH. The only active mycobacterial IMPDH orthologue, encoded by the *guaB2* gene, has been biochemically characterized[12,13]; however, most of these studies were limited to the enzyme with the deleted CBS domain (ΔCBS)[14]. Consequently, there is a lack of experimental evidence for the allosteric regulation of mycobacterial IMPDH.

In this study, we show that IMPDH from *Mycobacterium smegmatis* (*Msm*IMPDH) and *Mycobacterium tuberculosis* (*Mtb*IMPDH) are inhibited by a combination of ATP and either GTP or (p)ppGpp. Based on the series of cryogenic electron microscopy (cryo-EM) structures of *Msm*IMPDH in complex with its allosteric regulators and substrates in different functional states, we propose an integrated molecular model for the allosteric regulation of mycobacterial IMPDH. Our model explains how the inhibitory combination of ATP and guanine nucleotides locks the IMPDH octamer into an inactive compacted form, preventing the binding of the IMP substrate. Additionally, we show that IMP binding drives the expansion of octameric assemblies through extensive rearrangements of the active site environment.

## Results

### *Msm*IMPDH activity is allosterically regulated by both GTP and (p)ppGpp in an ATP-dependent manner

To test the response of *Msm*IMPDH to the purine effectors, we first conducted a biochemical characterization of the *Msm*IMPDH enzyme. The kinetic analysis of *Msm*IMPDH revealed that the enzyme exhibits Hill kinetics with respect to the IMP substrate ($K_{0.5, IMP} = 139 \pm 4 \, \mu M$; $n_H = 2.1 \pm 0.1$), and Michaelis–Menten kinetics with respect to NAD+ cofactor ($K_{m, NAD^+} = 800 \pm 17 \, \mu M$), with apparent inhibition at high NAD+ concentrations (Supplementary Fig. 2 and Supplementary Table 1). Furthermore, *Msm*IMPDH was not inhibited by ATP, GTP, or ppGpp when present separately (Supplementary Fig. 3 and Supplementary Table 2). However, GTP or ppGpp, when added in combination with ATP, exhibited strong K-type allosteric inhibition (Fig. 1a, c,

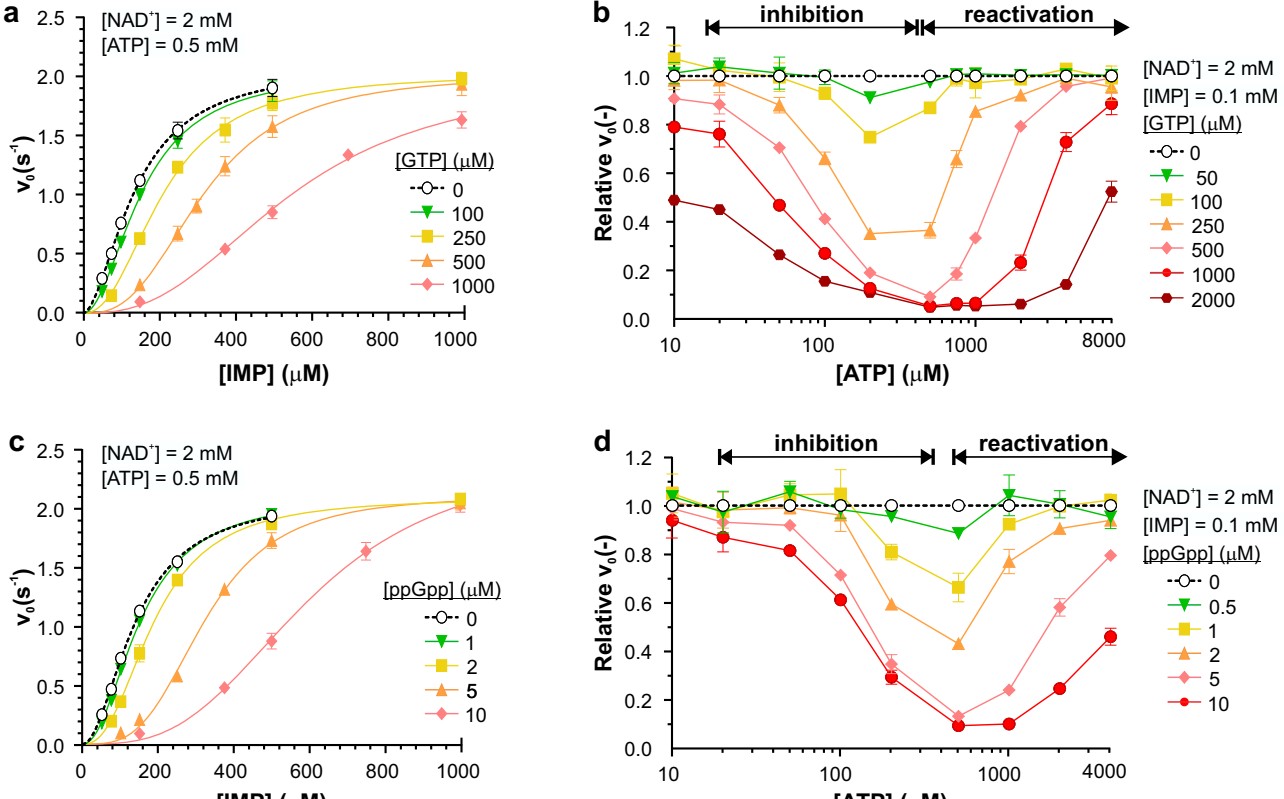

**Fig. 1 | *Msm*IMPDH is regulated by both GTP and ppGpp in an ATP-dependent manner. a, c** Initial velocity, fitted with the Hill equation, is plotted as a function of IMP concentration in the presence of 500 μM ATP and varied concentrations of (**a**) GTP and (**c**) ppGpp, respectively (*n* = 3). Control kinetics in the presence of 500 μM ATP only are shown in black dashed line. **b, d** Relative velocity is plotted as a function of the ATP concentration at set concentrations of (**b**) GTP and (**d**) ppGpp (*n* = 2). The substrates IMP and NAD+ were fixed at concentrations of 100 μM and

2 mM, respectively. The relative velocity value was calculated as the ratio of the initial velocity of the reaction in the presence of ATP and GTP or ATP and ppGpp to the control reaction containing only the substrates and the corresponding ATP concentration. Specific ATP concentration ranges above the graphs indicate where inhibition and reactivation occur. All data are presented as mean values with error bars representing the SD.

Supplementary Fig. 4, and Supplementary Table 2). Notably, $Mg^{2+}$ ions are required for the full inhibitory effect of these effectors (Supplementary Fig. 5).

Next, we compared the inhibitory effect of either GTP or ppGpp (at a set concentration range) as a function of changing ATP concentration, while maintaining a fixed sub-saturation concentration of IMP substrate and NAD$^+$ co-factor. In general, individual plots of relative velocity versus ATP concentration at a set concentration of GTP or ppGpp (Fig. 1b, d) followed a U-shaped trend. The left part of the U-shaped plot indicates that a certain ratio of ATP is necessary for effective inhibition of *Msm*IMPDH by GTP or ppGpp. The bottom of the U-shape indicates the concentration ratio between ATP and GTP or ppGpp for the maximum inhibitory effect. The right part of the plot shows the reactivation of *Msm*IMPDH at higher concentrations of ATP, thus overcoming the inhibitory effect of GTP or ppGpp. Notably, ppGpp and its related pppGpp and pGpp nucleotides were two orders of magnitude more potent *Msm*IMPDH inhibitors than GTP (Supplementary Fig. 6 and Supplementary Table 3). Importantly, the fact that the presence of ppGpp strengthened the inhibitory effect of GTP on *Msm*IMPDH indicates a functional interplay of those effectors (Supplementary Fig. 7). In addition, we tested the inhibitory potential of several bacterial purine signalling molecules, demonstrating that ppApp and Ap4G can also act as inhibitors of *Msm*IMPDH activity (Supplementary Fig. 8).

To validate the compatibility of our *Msm*IMPDH model with IMPDH from *M. tuberculosis*, we isolated and biochemically characterized *Mtb*IMPDH. Our experiments revealed highly similar catalytic properties between *Mtb*IMPDH and *Msm*IMPDH, including the cooperative behaviour of IMP ($n_H = 1.68 \pm 0.06$) and comparable response to ATP and GTP/ppGpp nucleotides (Supplementary Figs. 9, 10 and

Supplementary Table 4). In summary, our biochemical data indicate that the ratio of ATP to either GTP or ppGpp, along with IMP concentration, regulates the reaction rates of both *Msm*IMPDH and *Mtb*IMPDH. This finding contrasts with other reported prokaryotic IMPDHs, which typically demonstrate selectivity for either GTP or (p) ppGpp as the inhibitory molecule[2].

## *Msm*IMPDH octameric organization fluctuates between extended and compressed forms depending on the bound allosteric effector and substrate

To explore the mechanism of *Msm*IMPDH response to adenine and guanine nucleotides, we employed structural analysis using single-particle cryo-EM. We collected five experimental datasets of IMPDH with different combinations of IMP substrate and the allosteric effectors ATP, GTP, and ppGpp (Supplementary Table 5). Using cryo-EM data at a resolution of 2.4–3.1 Å, we constructed nine atomic models of *Msm*IMPDH (Supplementary Fig. 11). Similar to previously reported structures of IMPDHs, all of the structures determined in our study shared a conserved two-domain architecture consisting of catalytic and CBS domains. In all of the structures we observed, *Msm*IMPDH adopted an octameric assembly composed of a dimer of tetramers (Fig. 2a), with the exception of the apo dataset where a minor portion of tetrameric particles was present (Supplementary Fig. EM 1). In this conformation, catalytic domains form a central barrel-like structure flanked by CBS domains, which interact in pairs to link the opposite protomers. The ATP, GTP, and ppGpp effectors were present in the binding sites of CBS domains, which correlates with the conserved binding sites in the CBS domains of other IMPDHs. The highest structural variability of all the structures was observed in the relative positioning of the CBS domains and the catalytic core tetramers.

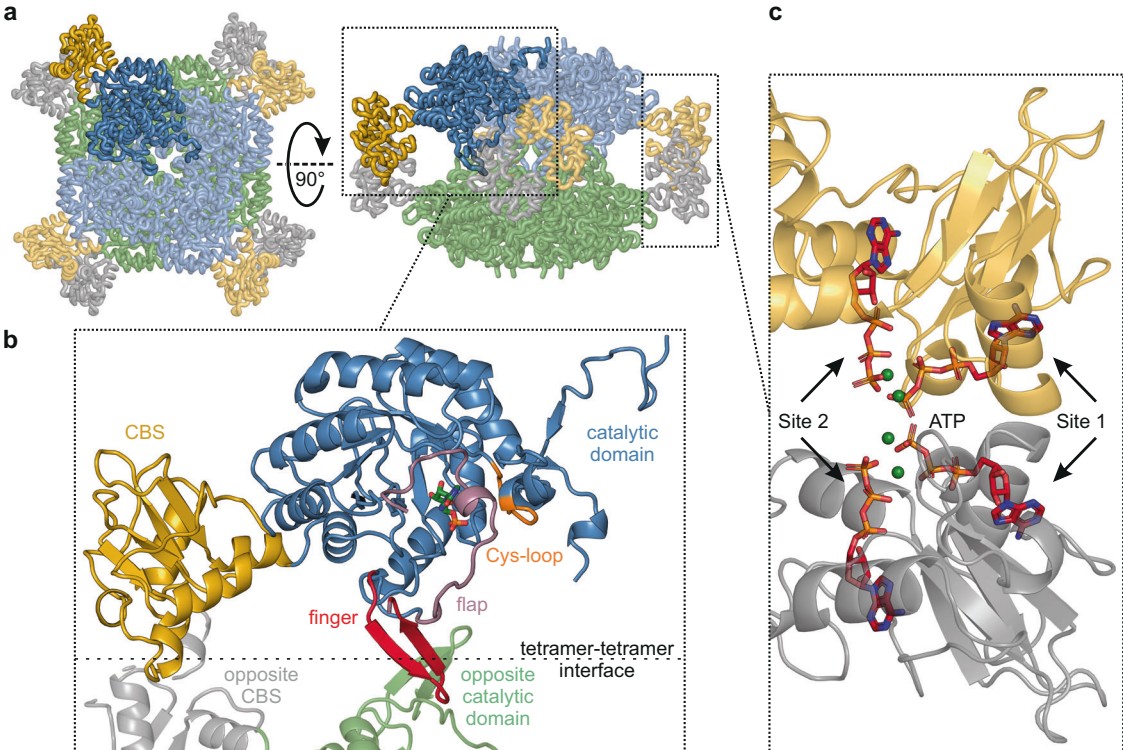

**Fig. 2 | Architecture of *Msm*IMPDH. a** Side and top views of an *Msm*IMPDH octamer: the first tetramer is depicted in blue and gold, with its opposite tetramer in green and grey; the *Msm*IMPDH monomer is highlighted in saturated colours. **b** Structural details of the highlighted *Msm*IMPDH monomer. The catalytic domain is depicted in blue and the CBS domain in gold. The flexible loops of the Cys-loop (residues 320–326) are shown in orange, the finger (residues 391–404) in red, the flap (residues 405–450) in purple, and the IMP molecule in green. The two fingers and distal parts of the flaps form a binding interface between the two opposite protomers. **c** Four ATP molecules (in red) bound within Site 1 and 2 in a complex with four $Mg^{2+}$ ions (in green) form an interface between two neighbouring CBS domains.

Additionally, we noted flexibility in the mobile loops of the catalytic domain itself, including the Cys-loop (residues 320–326, using *Msm*IMPDH residue numbering unless stated otherwise), the finger (residues 391–404), and the flap (residues 405–450). Importantly, the finger and the flap are also constituents of the interface between the two core tetramers (Fig. 2b).

The determined structures of *Msm*IMPDH can be divided into two conformations: extended octamers and compressed octamers. These conformations differ by a 10–16 Å inter-tetrameric distance between the most extended and compressed states. (Supplementary Fig. 12). The structures also differ in the degree of rotation between the opposite tetramers. The static structures represent the most defined states within the dynamically changing *Msm*IMPDH octameric assembly, which can fluctuate between extended, compressed and mutually rotated forms. The dynamic nature of this assembly is illustrated by the heterogeneity in the 3D classifications of the cryo-EM datasets (Supplementary Fig. 11 and Supplementary Figs. EM 1–14).

### ATP stabilizes CBS-domain interaction without inducing *Msm*IMPDH octamer extension

Several studies[11,15,16] have linked the extended active octamer conformation to ATP binding. Surprisingly, in our study, the addition of ATP under cryo-EM conditions failed to induce the extended conformation of the *Msm*IMPDH octamers. The datasets of *Msm*IMPDH apo and ATP-bound states were homogeneous, yielding only one major 3D class (Supplementary Figs. EM-2, 4). Both structures adopt highly similar compressed octamer conformation, stabilized by the interactions of the finger and flap loops within the intra-tetrameric interface.

The most notable effect of ATP on the *Msm*IMPDH structure was the stabilization of the CBS domain pairs. While the CBS domains in the apo form were not resolved at all, the CBS domains in the ATP-bound form established a defined position with respect to the catalytic core (Supplementary Fig. 11). One CBS domain pair exhibited interpretable density, enabling us to construct an atomic model, while the other three had only low-resolution contours. The defined CBS domain pair has a clear density for the ATP molecules in canonical Site 1 and Site 2. The β- and γ-phosphates of both ATP molecules jointly coordinate two magnesium ions, forming an interface with the neighbouring CBS domain, thus facilitating the formation of a stable CBS domain pair (Fig. 2c). The ATP molecule in Site 1 forms the core of this interface and contributes to most of the interactions between the two opposing CBS domains. Importantly, the mutations of residues involved in ATP-binding Site 1 (R157A, D158A, E200A, and K201A) did not affect activity while slightly decreasing $K_{0.5, IMP}$ and $n_H$ parameters, but resulted in the loss of *Msm*IMPDH inhibition by GTP and ppGpp (Supplementary Fig. 13 and Supplementary Table 6). We confirmed the significantly reduced ATP binding in these mutants using the differential radial capillary action of ligand assay (DRaCALA) technique (Supplementary Fig. 14a). Interestingly, some of the mutations may have affected the affinity of CBS domain pairs in the apo form, as evidenced by changes in the oligomeric state of MsmIMPDH (Supplementary Fig. 14b). These results confirm that the CBS domain interaction, facilitated by the binding of ATP to Site 1, is crucial for the allosteric regulation of *Msm*IMPDH.

To investigate the overall dynamics of the *Msm*IMPDH octamers, we employed hydrogen/deuterium exchange–mass spectrometry (HDX–MS). The apo *Msm*IMPDH showed a notably increased deuterium uptake in the region of the flap loop, the C-terminus, and the CBS domain relative to the other parts of the protein (Fig. 3a). The highly dynamic nature of these regions correlates with the drop we observed in the local resolution of the same regions and the absence of the CBS domain in the cryo-EM map in the apo form. In contrast, the presence of ATP significantly decreased deuterium uptake in the region of the CBS domains compared to the apo form (Fig. 3b). This decrease is

consistent with the combined effects of ATP binding and subsequent dimerization of the CBS domains and their limited mobility. Taken together, ATP stabilizes the interaction of the CBS domains without inducing the extension of the *Msm*IMPDH octamers.

### IMP induces *Msm*IMPDH octamer extension

Next, we searched for the conditions leading to *Msm*IMPDH octamer expansion. The only extended *Msm*IMPDH structure was obtained from samples containing ATP in combination with the substrate IMP. The dataset of *Msm*IMPDH in the presence of ATP and IMP was highly heterogeneous, resulting in multiple 3D classes of *Msm*IMPDH octameric assemblies differing in their compression and extension range (Supplementary Fig. EM 6). This illustrates the overall highly dynamic nature of the ATP- and IMP-bound complex. Three distinct classes were refined: extended, compressed, and intermediate. The compressed structure closely resembled the compressed forms of apo and ATP-bound *Msm*IMPDH. The IMP molecule was only partially present, represented by weak densities observed for its phosphate and sugar moieties. In contrast, IMP was clearly visible in the intermediate and extended structures.

Based on our detailed comparison of the compressed and extended *Msm*IMPDH octamers, IMP binding triggered extensive reorganization of the mobile Cys-, finger-, and flap loops near the active site and, concomitantly, of the tetramerization interface. The binding mode of IMP to *Msm*IMPDH is consistent with previously published structures of both prokaryotic and eukaryotic IMPDHs[3,17–20]. In all our compressed structures, the Cys-loop, containing the catalytic Cys325 residue, adopted the inactive open conformation, but in the extended form, the active closed conformation prevailed, correlating with the presence of IMP (Fig. 4a, d). Furthermore, IMP binding facilitated the ordering of the first part of the flap (residues 405–415). This created interactions between the Met408 and Gly409 backbone amides, forming contacts with the IMP base. Additionally, the hydroxyl group of Tyr405 established contact with the phosphate group of IMP (Fig. 4b, d). Those contacts were absent in the compressed form. This IMP-induced rearrangement of the flap loop forced the directly connected finger loop (residues 391–404) to move by about 3.5 Å between the compressed and extended conformations (Fig. 4c, d). Finally, the displacement of the finger loop drives loosening of the tetramer dimerization interface, leading to the disassembly and eventual extension of the compressed octamer. In their extended form, the tetramers maintained only very limited contacts at the tips of the finger loops, whereas the octamers were mainly held together by the interaction of the CBS domains (Fig. 4e).

IMP alone, in the absence of any other effector, led to the disassembly of the majority of *Msm*IMPDH octamers into tetramers under cryo-EM conditions (Supplementary Figs. EM 11, 12). In contrast, ATP alone, even at a high concentration of 10 mM, did not result in any extension (Supplementary Figs. EM 13, 14). Our analysis of the rearrangement of the overall structure using small angle X-ray scattering (SAXS) at 56 μM *Msm*IMPDH concentration (Fig. 4f and Supplementary Figs. SAXS 1–5) confirmed that the *Msm*IMPDH apo form primarily favoured the compressed octameric conformation, which was also the case in the presence of ATP and GTP. IMP alone induced tetramerization, and an extended octameric conformation when combined with ATP. Our mass photometry (MP) analysis (Fig. 4g) revealed that the apo form of *Msm*IMPDH at 20 nM concentration shows a mixture with slight preference for octamers. While the addition of ATP shifted this equilibrium heavily towards the octameric form, the addition of IMP tipped the balance towards the tetrameric arrangement. These observations align well with our cryo-EM and SAXS data. The differences in the tetramer:octamer equilibrium observed in the apo form of *Msm*IMPDH by mass photometry and SAXS measurements are likely caused by the dramatically different

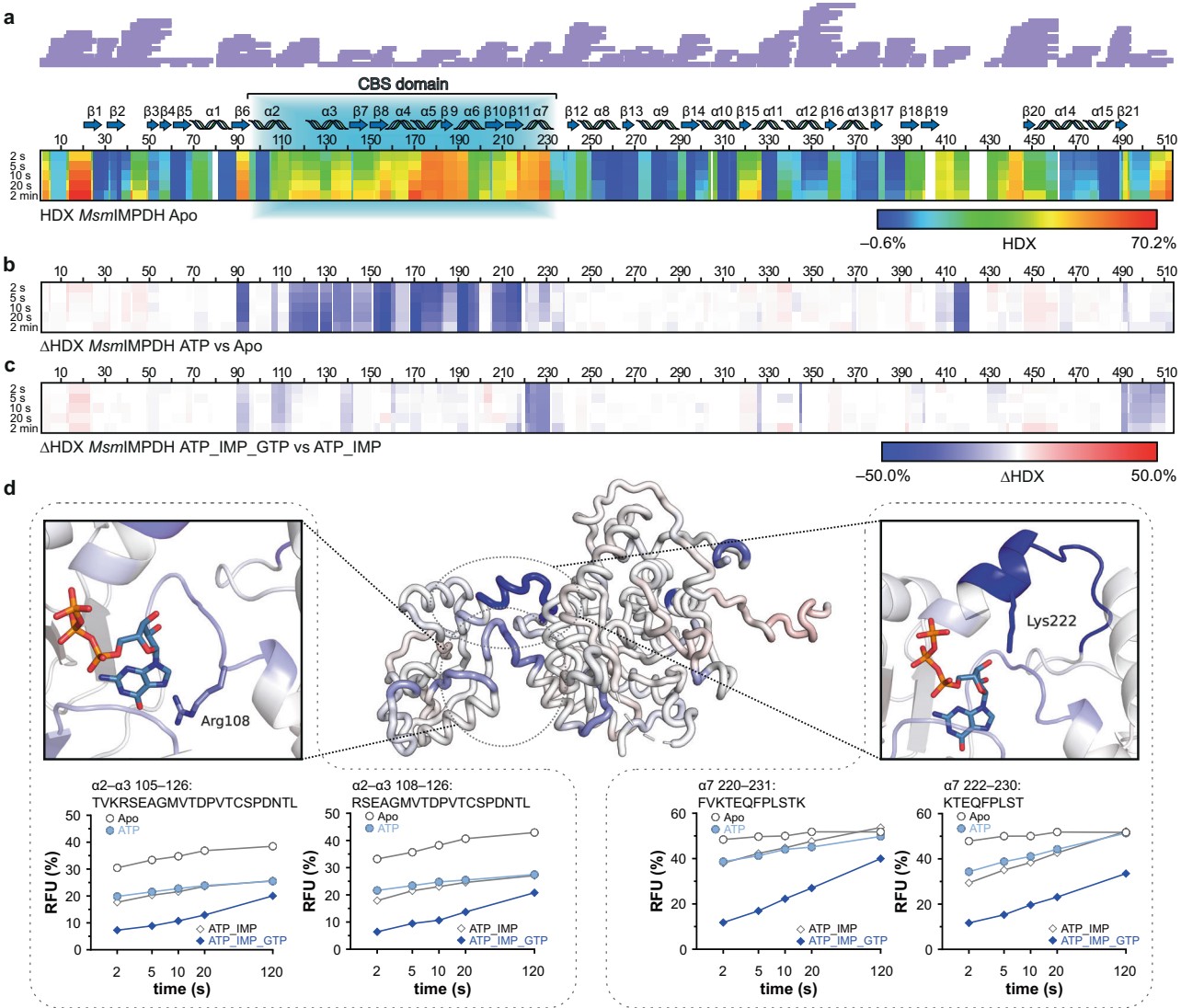

**Fig. 3 | HDX–MS profiles of *Msm*IMPDH showing a decrease in the dynamics of the CBS domain upon binding of ATP and GTP. a** Heat map of apo *Msm*IMPDH (constructed from 221 generated peptides) shows the deuteration levels of each amino acid residue in *Msm*IMPDH obtained at 4 °C over a time course of 2–120 s. Each line represents one time point (2 s, 5 s, 10 s, 20 s, and 2 min). The degree of deuteration is represented by the relative fractional uptake, scaled in rainbow colours from the minimum to the maximum observed uptake. The residues with no sequence coverage are shown as white gaps in the heat map. The sequence coverage is depicted by purple lines above the sequence. Differential heat maps of *Msm*IMPDH under conditions of (**b**) apo vs ATP, and (**c**) ATP–IMP vs ATP–GTP–IMP. Differences in the relative fractional uptake are displayed in a blue–white–red colour scale: residues with decreased accessibility are in blue, and residues with increased HDX are in red. **d** The ΔHDX changes of ATP–IMP vs ATP–GTP–IMP at 10 s are mapped on the *Msm*IMPDH monomer structure. Red indicates elevated deuteration, and blue less extensive deuteration of the given region. Hinge regions with residues involved in the binding of GTP (Arg108 and Lys222) are outlined by separate black frames. Deuterium uptake plots show the evolution in the deuteration of four representative peptides in the hinge region. In the graphs, each line represents one protein state, and each point represents one time point over a time course of 2–120 s. For comparison, the protein states are displayed by the same symbols. The drops observed in the HDX rate are displayed in light- and dark-blue colours.

protein concentrations used in both methods. All things considered, these results clearly demonstrate that the extension dynamics of *Msm*IMPDH is driven by the binding of IMP and not by ATP, as previously assumed for other IMPDHs.

### Binding of either GTP or ppGpp locks *Msm*IMPDH octamers in a compressed inhibited conformation

To mechanistically elucidate the *Msm*IMPDH inhibition, we analyzed the structures of complexes of *Msm*IMPDH with ATP:GTP or ATP:ppGpp in the presence of IMP. The dataset for *Msm*IMPDH with ATP and GTP yielded two major compressed 3D classes at overall resolutions of 2.35 and 2.43 Å, respectively. Both structures share a remarkably similar overall conformation, differing only slightly by about 3 Å in compression. They are also characterized by well-resolved CBS domain pairs, with a clear density for ATP bound to canonical Site 1 and for GTP at canonical Site 2. The *Msm*IMPDH dataset collected in the presence of ATP and ppGpp yielded two compressed structures at resolutions of 2.73 and 2.76 Å, respectively, differing slightly in the degree of octamer compression. Both ppGpp structures also exhibited well-defined density for the CBS domains, along with one molecule of ATP at canonical Site 1 and one ppGpp molecule in the ppGpp-dedicated binding pocket (distinct from Site 2). The general binding modes of all the ligands (ATP, GTP, and ppGpp) are remarkably similar to those of other reported IMPDHs. The ppGpp binding site partially overlaps with canonical Site 2. The δ- and ε-phosphates of ppGpp coordinate two $Mg^{2+}$ ions in a similar manner to the β- and

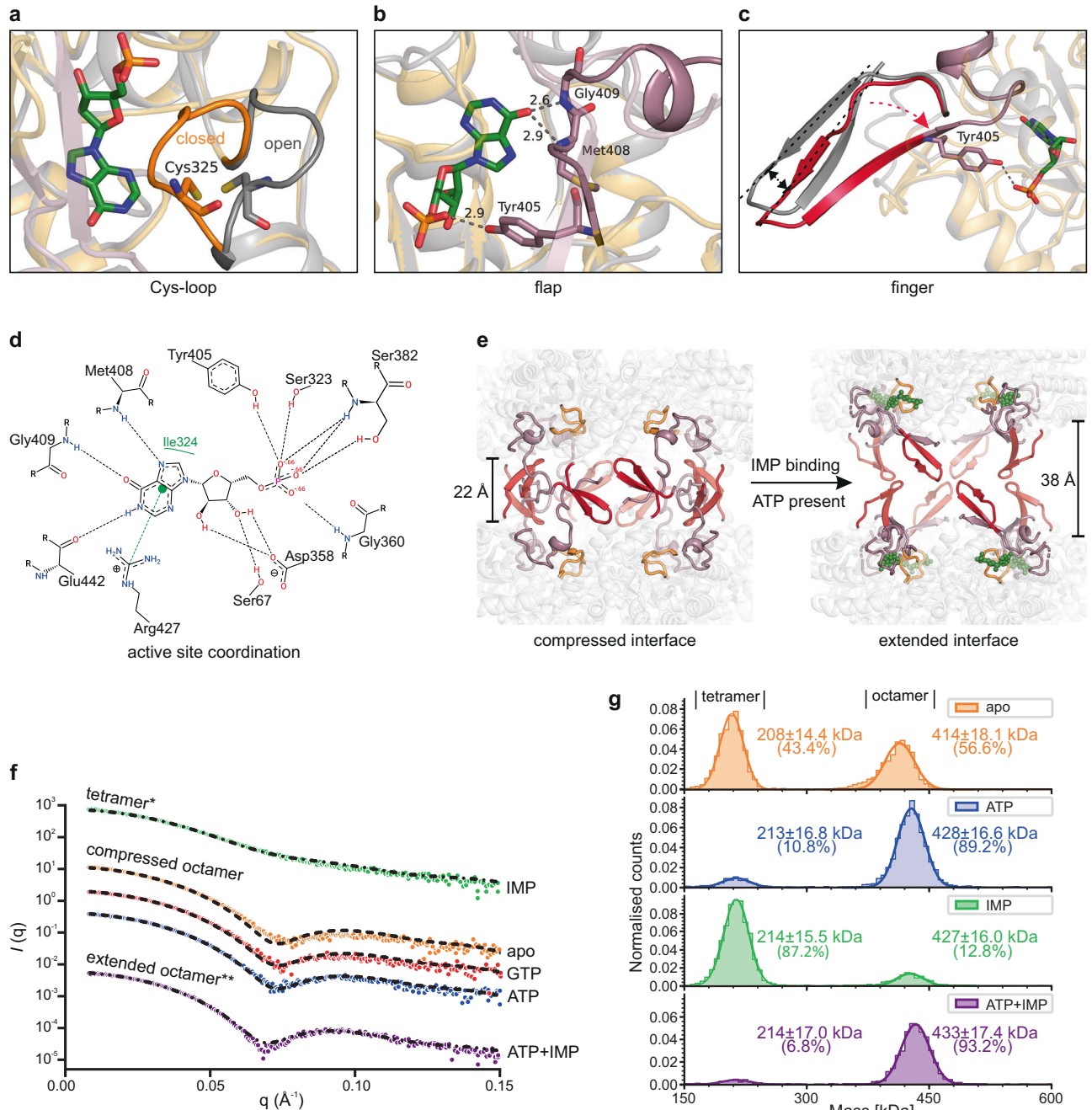

**Fig. 4 | IMP binding induces reorganization of the *Msm*IMPDH active site and subsequent dissolution of the tetramer dimerization interface. a** The binding of IMP (in green) induces movement of the Cys-loop (residues 320–326, in orange) from an open conformation to a closed conformation. **b** The initial part of the flap loop (residues 405–415, in purple) interacts with IMP through residues Tyr405, Met408, and Gly409. The contacts are depicted by the grey dotted lines; each number indicates the distance in Å. **c** IMP binding induces a movement of about 3.5 Å of the entire finger loop (residues 391–404 in red), as indicated by the red arrows. **d** The two-dimensional representation shows the contacts between IMP and residues forming the *Msm*IMPDH active site. **e** Changes to the finger (in red), flap (in purple), and Cys-loop (in orange) after IMP binding (green spheres) lead to extensive rearrangement of the tetramer–tetramer interface, enabling octamer expansion. For consistency, the ATP-bound *Msm*IMPDH structure was chosen as the model for the compressed conformation across all panels (grey cartoon in **a**–**c**).

**f** SAXS profiles of 56 μM *Msm*IMPDH show the effect of nucleotides on the quaternary structure. For ease of visualization, plots are conveniently displaced along the *y* axis to represent the three obtained conformations: (green) tetramers induced by 10 mM IMP; (orange) compressed octamers in the apo state, (red) with 10 mM GTP, and (blue) with 10 mM ATP; and (purple) extended octamers induced by 10 mM ATP and IMP. Dashed lines show the theoretical SAXS profiles calculated from the respective cryo-EM structures fitted to the experimental scattering curves. The curve fitted to the IMP dataset (marked by *) represents a 75:25% mixture of tetramers and octamers, while the fit for the ATP + IMP dataset (marked by **) indicates a 38:62% mixture of compressed and extended conformations, as calculated using the OLIGOMER program[53]. **g** Mass photometry profiles of 20 nM *Msm*IMPDH in its apo state and in the presence of 5 mM ATP and/or IMP reveal two distinct peaks. The observed particle masses of the first and second peaks correspond with the *Msm*IMPDH tetramer (213.2 kDa) and octamer (426.4 kDa).

γ-phosphates of ATP and GTP bound to Site 2. For these steric reasons, the binding of ppGpp to either ATP or GTP at Site 2 is mutually exclusive (Supplementary Fig. 15).

Intriguingly, the more compressed forms of GTP- and ppGpp-bound octamers (Supplementary Fig. 16) feature a swap in the flap loops. Instead of the regular *cis* interaction of the flap loop with the active site of the resident protein chain protomer, the flap loops interact in *trans* with the active sites of the opposite protomers. This *trans* interaction of the opposite protomers further compacts the

catalytic core tetramerization interface. However, the existence of such a 'super-compressed' octameric assembly has not been reported before, and its functional significance remains to be determined.

Next, we compared the differences in the Site 2 binding mode of ATP compared to GTP, and their impact on the conformation of the whole *Msm*IMPDH octamer. The ATP base is bound in *anti* conformation in respect to the ribose, while the GTP base is bound in *syn* conformation (Fig. 5a, b). The amino group of ATP is specifically recognized by hydrogen bonding with the backbone carbonyl of

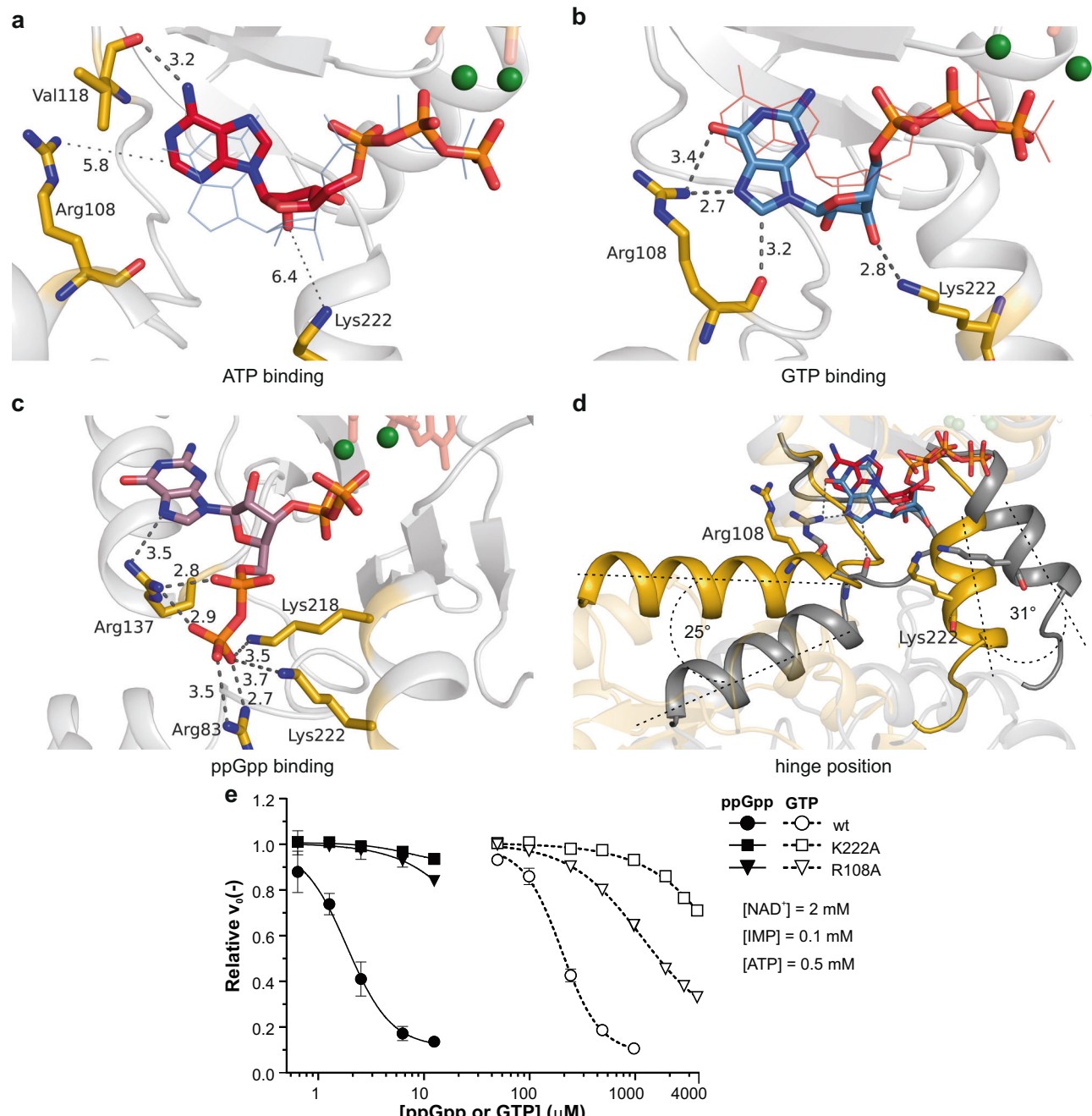

**Fig. 5 | Differential modes of binding of ATP, GTP, and ppGpp to the CBS domain of *Msm*IMPDH impact the mobility of the hinge regions. a** ATP binds at Site 2 with the base in *anti* conformation relative to the ribose. **b** The GTP molecule binds with the base in *syn* conformation. The other nucleotide is shown as thin lines in the background for comparison of a and b. The contacts with Arg108 and Lys222 (in gold) are depicted by grey dotted lines; each number indicates the distance in Å. **c** The ppGpp molecule binds at a separate binding site in the CBS domain. **d** The hinge regions are flexible in the case of ATP binding (in gold), while the GTP locks the hinges in a fixed position (in grey). The side chains of Arg108 and Lys222 are depicted as sticks. **e** The mutation of Arg108A and Lys222A strongly reduces the inhibition of *Msm*IMPDH by ppGpp and GTP. NAD⁺ and IMP substrates were fixed at concentrations of 2 mM and 100 μM, respectively; ATP was fixed at 500 μM. The relative velocity value was calculated as the ratio of the initial velocity of the reaction to the control reaction of the corresponding enzyme (n = 3). Data are presented as mean values with error bars representing the SD.

Val118. In contrast, the *syn* conformation of GTP induces tilting of the ribose ring and subsequent reorganization of the environment in close vicinity of Site 2. Most notably, the side chain of Arg108 specifically recognizes the 6-oxo group of the GTP, while the side chain of Lys222 forms contacts with the ribose 3'-hydroxyl group (Fig. 5b). These interactions were not observed in the case of ATP binding (Fig. 5a).

Both Arg108 and Lys222 are part of the linker regions between the CBS domain and its respective catalytic domain. The linker regions of residues 108–116 and 216–228 act as two flexible hinges through which the opposite tetramers perform movements of extension and compaction. The fixation of Arg108 and Lys222 to GTP induces a locked orientation of the CBS domains relative to the catalytic core (Fig. 5d). This orientation is further stabilized by notably increased bonding between Arg and Lys residues along with acidic counterparts located around the regions connecting the CBS and catalytic domains. Several of these residues, such as Arg83, Lys231, and Arg236, protrude between the two domains, facilitating mutual contacts. Overall, this intertwined network, ordered by the binding of GTP to Site 2, forms a latch that stabilizes the compressed form of the *Msm*IMPDH octamer. Although ppGpp binding induces a similar mutual orientation of the CBS and catalytic domains, it also involves interactions of the side chains of residues Arg83, Arg137, Lys218, and Lys222 with the β- and γ-phosphate groups (not present in either ATP or GTP) of ppGpp (Fig. 5c). The critical importance of the Lys222 and Arg108 residues is reflected in a substantial reduction in the sensitivity of *Msm*IMPDH to GTP and ppGpp inhibition of the corresponding alanine substitution mutants (Fig. 5e, Supplementary Fig. 17, and Supplementary Table 6).

To investigate the mobility of the CBS domains locked by the combination of ATP and GTP, we compared HDX–MS profiles of the inhibited *Msm*IMPDH with the active form bound to ATP and IMP. The deuteration profiles exhibited a comparable level of deuteration across the CBS domains, except for a significant stabilization observed in both hinge regions in the presence of GTP. (Fig. 3c). The stabilization displayed by peptides 105–126 and 108–126 in the first hinge and peptides 220–231 and 222–230 in the second hinge reflected a consistent decrease in the HDX rate (Fig. 3d).

Collectively, our results describe two unique and yet similar ways in which the binding of GTP or ppGpp to their distinct sites stabilizes the *Msm*IMPDH octamer in its compressed form. The binding of GTP or ppGpp fixes the CBS domain in a locked position relative to the catalytic core through interaction with the dedicated hinge regions. The hinges are folded in a way that leads to compression of the octameric assembly, which consequently favours the finger and flap loops to form the stable catalytic core tetramer–tetramer interface. This configuration prevents IMP binding, ultimately resulting in enzyme inhibition.

## Discussion

IMPDH is recognized as a critical regulator of ATP and GTP levels, the maintenance of which is of paramount importance for bacterial survival[21–23]. The data presented here not only reveal the mechanistic details of the allosteric regulation of mycobacterial IMPDH, but also have two major implications for the model currently proposed for other bacterial IMPDHs[2]. These implications are (i) that the distinct mechanism of active, extended IMPDH octamer formation is driven by the binding of IMP, not by ATP and (ii) that, contrary to previous suggestions, the enzyme is in fact sensitive to inhibition by both GTP and/or (p)ppGpp nucleotides.

Based on our structural and biophysical analyses, we propose an integrated model for the regulation of mycobacterial IMPDH (Fig. 6 and Supplementary Movie 1). We demonstrate that in the apo form of *Msm*IMPDH, the flap and finger loops, which are constituents of the active site of the catalytic domains, form a packed tetramer–tetramer interface, which brings the two tetramers close together. Thus, the two tetramers adopt predominantly a compressed inactive octameric

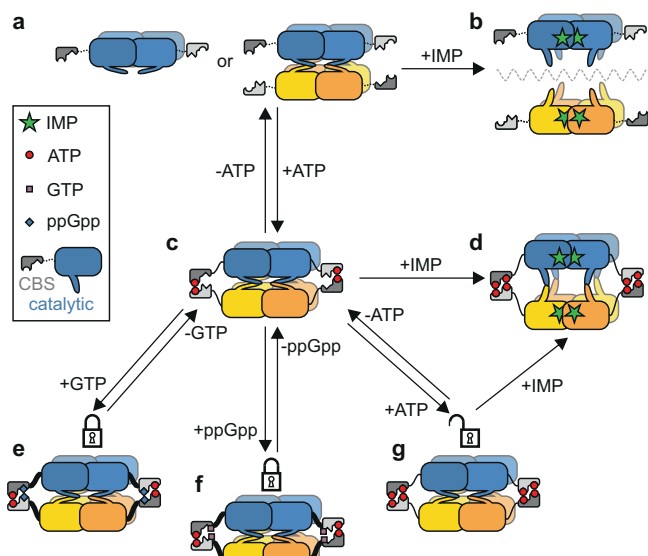

**Fig. 6 | Schematic model of *Msm*IMPDH allosteric regulation illustrating the interplay of nucleotide interactions and their impact on the activity and conformation of *Msm*IMPDH.** The top tetramer is depicted in blue, its opposite tetramer in gold, and the CBS domains in shades of grey. **a** The CBS domains are free in the apo form that exists in both tetrameric and octameric conformations. **c** The binding of ATP to Site 1 stabilizes the dimers of the CBS domains and thus the octameric conformation. **b, d** IMP binding drives the formation of the active tetramers (**b**) or extended octamers stabilized by ATP (**d**). **e, f** This extension is blocked by the binding of GTP (**e**) or ppGpp (**f**), thus locking the hinge regions that link the CBS and catalytic domains (depicted as thick black linkers). **g** Binding of ATP to Site 2 competes with GTP and ppGpp, while enabling the IMP-induced extension.

conformation (Fig. 6a), that reduces the affinity for substrate IMP. Furthermore, our cryo-EM and SAXS data show that the presence of IMP drives the enzyme towards the extended conformation state, forcing both loops in the direction of the active site. The mode of IMP binding observed in the active site is consistent with previously published crystal structures of other bacterial IMPDHs, including that of *M. tuberculosis* ΔCBS IMPDH[5,14,24]. Therefore, the reorganization of the flap and finger loops towards the active site ultimately serves to loosen the tetramer–tetramer interface, thereby enabling extension of the octameric assembly and enzyme activity (Fig. 6b, d).

We show that, like other prokaryotic IMPDHs[11,15], *Msm*IMPDH's canonical Site 1 exclusively binds ATP, facilitating dimerization of neighbouring CBS domains (Fig. 6c). This ATP-driven interaction is crucial for *Msm*IMPDH inhibition, mediated by GTP and (p)ppGpp through canonical Site 2 and the dedicated (p)ppGpp binding site, respectively. The binding of GTP or ppGpp to their respective sites coordinates the Arg108 and Lys222 residues within the hinge regions between the CBS and catalytic domains in a way that locks the octamer in a compressed conformation (Fig. 6e, f). This GTP- or ppGpp-stabilized octamer compression compels formation of the flap and finger loops at the tetramer–tetramer interface, thereby preventing the coordination of IMP by the loops in the active site and ultimately inhibiting the enzyme.

In contrast to GTP or ppGpp, ATP binding to *Msm*IMPDH Site 2, which closely resembles that of other prokaryotic IMPDHs[15], does not involve the coordination of basic residues in the hinge regions. Consequently, this facilitates a flexible connection between the CBS and catalytic domains (Fig. 6g). While this flexible connection allows for octamer extension, our structural data show that ATP binding to Site 2 itself is not the driving force of the expansion. Instead, we propose that the primary function of the binding of ATP to Site 2 is direct steric

competition with the binding of guanine nucleotides. In this way, the enzyme senses the actual ratio of the ATP and guanine nucleotides, which directly and mechanistically translates into conditions that either permit (Fig. 6g) or restrict (Fig. 6e, f) IMP-driven expansion of the octamer.

Recent studies have shed light on the allosteric regulation of human IMPDH1 and IMPDH2 isoforms[3,25,26]. Although the molecular details of the regulation differ from those observed for MsmIMPDH, the general mechanism is conserved: adenine nucleotides stabilize extended octamers, whereas guanine nucleotides inhibit enzyme activity by locking a compressed conformation. In addition, human IMPDHs have another level of regulation, oligomerizing into filaments called cytoophidia in response to purine metabolic states. This filamentation, unique to vertebrate IMPDHs, involves either a flat or bowed conformational change during filament assembly, characterized by an apparent tilt of the protomers relative to the fourfold symmetry axis of the tetramer[25]. While all of our MsmIMPDH structures are not directly comparable to the flat or bowed conformations seen in eukaryotic IMPDHs, they also show some degree of flexibility within the tetramer, reminiscent of the flat and bowed conformational transitions (Supplementary Fig. 18). Nonetheless, the relative range of those movements is comparatively limited. This flexibility in non-filamentous IMPDH has been previously noticed by Buey et al.[2], however, its functional relevance remains unclear.

At the structural level, all reported IMPDHs, including the MsmIMPDH structures presented here, show remarkable similarity in their compacted and extended conformations. In eukaryotic IMPDHs, inhibition is mediated by the binding of two GTP molecules, one to canonical Site 2 and the other to eukaryote-specific Site 3. While the binding of GTP at Site 2 is functionally conserved with that of MsmIMPDH, the mode of GTP binding at Site 3 differs. Interestingly, Site 3 binding also involves interactions with basic residues in the hinge regions. Although these residues differ from those involved in MsmIMPDH ppGpp binding, both bacterial and eukaryotic IMPDHs appear to leverage a previously undescribed common molecular mechanism to lock the hinge regions (Supplementary Fig. 19).

Aside from the mechanism itself, we demonstrated that MsmIMPDH is inhibited by both GTP and (p)ppGpp. This observation contrasts with the hypothesis[2] that the IMPDHs of most bacterial phyla, including Actinobacteria, do not bind GTP at canonical Site 2. In contrast, our extensive phylogenetic analysis of IMPDH sequences in the Actinobacteria phylum revealed almost universal conservation of binding residues for both GTP and (p)ppGpp (Supplementary Fig. 20). Therefore, while further research is needed to elucidate species-specific differences, we hypothesize that the allosteric regulation of other actinobacterial IMPDHs might resemble the model established for the mycobacterial enzyme.

With the above data in mind, let us now consider how MsmIMPDH regulates purine nucleotides homeostasis. In exponentially growing mycobacteria, intracellular levels of ATP and GTP remain within the millimolar range[27], which exceeds their respective binding affinities to MsmIMPDH by a factor of about 10. This suggests that MsmIMPDH activity is regulated by ATP/GTP competition within their saturation range. Once MsmIMPDH detects an ATP:GTP imbalance, it can restore equilibrium by either activating itself to produce XMP for GTP synthesis or inhibiting itself to retain IMP for subsequent ATP synthesis.

In contrast, the low micromolar range affinity of ppGpp to MsmIMPDH is approximately two orders of magnitude lower than that of ATP and GTP, corresponding to the basal intracellular levels of ppGpp during steady-state growth[28,29]. The fact that ppGpp is capable of strengthening MsmIMPDH GTP-induced inhibition suggests that ppGpp levels act as a fine-tuning parameter of the steady-state purine metabolic flow and, consequently, of the ATP:GTP ratio in exponentially growing cells. Additionally, complete inhibition of MsmIMPDH activity can occur during acute stress responses, since (p)ppGpp

in vivo levels typically rise significantly above concentrations of 100 $\mu$M[29]. Taken together, MsmIMPDH can directly adjust ATP and GTP cellular levels in concert with (p)ppGpp levels, thereby acting as a regulatory hub for the complex in vivo interplay between cell physiology and purine metabolism.

Previous studies have identified other individual bacterial signalling molecules, such as dinucleoside polyphosphates, as potential candidate regulators of IMPDH activity[23,30]. Indeed, in our study, we found that ppApp and Ap4G inhibited MsmIMPDH in the range of hundreds of micromolar (Supplementary Fig. 8). However, it should be noted that in vivo concentrations reported in the literature are typically two to three orders of magnitude lower[31], which casts doubt on their biological relevance for IMPDH regulation.

The competitive inhibitors of NAD$^+$ or IMP binding have been extensively screened with the aim of identifying potential antimycobacterial compounds[6,13,14,19,32–34]. Despite these efforts, no inhibitor exploiting this mode of action has been approved for use as an antibacterial drug. An alternative strategy is to use allosteric inhibitors to target the CBS domain instead of the active site[35]. However, this approach has been hindered by limited insight into the details of allosteric regulation of bacterial IMPDHs. The mechanistic principles underlying MsmIMPDH allosteric regulation presented in this study offer a valuable framework for developing more effective allosteric inhibitors of mycobacterial IMPDH.

## Methods
### Construction of plasmids
All plasmids used in the study are listed in Supplementary Table 7.

The recombinant DNA constructs were generated as follows: Q5 High-Fidelity DNA Polymerase (New England Biolabs) was employed for PCR DNA amplification, restriction enzymes were provided by New England Biolabs, DNA fragments were ligated using the In-Fusion HD Cloning Kit (Takara), and plasmid amplification was conducted using Escherichia coli DH5α. The final constructs were validated by Sanger sequencing (Eurofins). All synthesized oligonucleotides, provided by Eurofins and Generi Biotech, are listed in Supplementary Table 8.

To construct the MsmIMPDH expression plasmid (pRSF-HisTEV-MsmGuaB2), the entire coding sequence was amplified utilizing the primer pair #1 + 2 and genomic DNA of the M. smegmatis MC$^2$ 155 strain as a template. The amplified fragment was then inserted into pRSF-HisTEV via the SacI site. Similarly, the expression plasmid for MtbIMPDH (pET24d-HS_MtbGuaB2) was prepared using primer pair #19 + 20 with genomic DNA from M. tuberculosis as a template to generate the insert. This amplicon was inserted into the pET24d-HS plasmid, which was linearized by PCR (primers #21 + 22).

The set of expression plasmids for MsmIMPDH Ala substitution mutants was constructed as follows: the entire plasmid was amplified by PCR using primers carrying the desired substitutions (oligonucleotide pairs #3-x as noted by the name). The wild-type expression plasmid (pRSF-HisTEV-MsmGuaB2) was used as a template. After PCR, the mixtures were treated with DpnI to remove the template DNA and then ligated using the In-Fusion kit.

### Expression of MsmIMPDH
The recombinant MsmIMPDH and its mutant variants in N-terminal fusion with a His6x tag flanked by a TEV protease cleavage site was produced in the E. coli BL21 RIL LOBSTR strain[36] using pRSF-HisTEV-MsmGuaB2 plasmids. A ZYM-505 expression medium[37] supplemented with 100 $\mu$g ml$^{-1}$ kanamycin was inoculated with an overnight culture to an initial OD$_{600}$ value of 0.05 and then cultivated at 37 °C until it reached an OD$_{600}$ value of 0.5. The cultivation temperature was then lowered to 18 °C and the expression induced by 1 mM IPTG. After 18 h of cultivation, the cells were harvested by centrifugation at 8,000 $g$ for 15 min; pellets were frozen and stored at −80 °C until further use.

## Purification of *Msm*IMPDH for structural analysis

The frozen cell pellets were thawed and resuspended in lysis buffer (200 mM $K_2HPO_4$, pH 8.0, 2 M KCl, 2.5 mM TCEP, 10 mM imidazole, 0.5% Triton X-100, 1 mM PMSF, and 0.1 mg ml$^{-1}$ lysozyme) by stirring at 4 °C for 60 min. The cells were lysed by sonication and the lysate cleared by centrifugation at 38,000 *g* for 30 min at 4 °C. The cleared supernatant was loaded onto the HisTrap 5 mL HP column pre-equilibrated with buffer A (200 mM $K_2HPO_4$, pH 8.0, 2 M KCl, 2.5 mM TCEP, 10 mM imidazole). The column was washed with 15 column volumes (CV) of buffer A, followed by an additional 10 CV of buffer A supplemented with 60 mM of imidazole to wash out non-specifically bound contaminants. The His-tagged proteins were then eluted with elution buffer (200 mM $K_2HPO_4$, pH 8.0, 200 mM KCl, 2.5 mM TCEP, 500 mM imidazole). To cleave the His-tag, TEV protease was added to a final concentration of 0.05–0.1 mg ml$^{-1}$ followed by incubation of the sample at room temperature for 1.5 h. Next, the sample was dialyzed overnight using a 6–8 kDa membrane (Repligen) against an excess of buffer B (10 mM $K_2HPO_4$, pH 8.0, 10 mM KCl, 2.5 mM TCEP, 10 mM imidazole). The dialyzed sample was loaded onto a HisTrap 5 mL HP column pre-equilibrated with buffer B to remove TEV protease, the cleaved tag, and the uncleaved protein. To remove co-purifying nucleic acids, the flow-through was loaded onto a HiTrap Q HP 5 ml ion exchange column equilibrated with buffer B. Following a 10-column volume wash with buffer B, bound proteins were eluted using a gradient of buffer C (10 mM $K_2HPO_4$, pH 8.0, 1 M KCl, 2.5 mM TCEP). The fractions most enriched with *Msm*IMPDH (eluting within a range of 200 to 300 mM KCl) were collected and further purified by size-exclusion chromatography using a HiLoad 26/600 Superdex 200 pg column in buffer D (200 mM $K_2HPO_4$, pH 8.0, 1 M KCl, 0.5 mM TCEP). The purified *Msm*IMPDH sample was then transferred to storage buffer (50 mM Tris-HCl, pH 8.0, 0.5 mM TCEP) using a HiPrep 26/10 desalting column and concentrated to a final 21 mg ml$^{-1}$ using an Amicon Ultra centrifugal filter device with a cutoff of 50 kDa. Finally, the protein preparation was aliquoted, flash-frozen in liquid nitrogen, and stored at −80 °C. The purity of *Msm*IMPDH was analyzed by SDS–PAGE, and the identity of the proteins was confirmed by mass spectrometry. All FPLC columns and Äkta equipment was manufactured by Cytiva; each chromatography step was performed at 4 °C.

## Purification of *Msm*IMPDH mutants for activity screening

The frozen cell pellets were thawed and resuspended in lysis buffer (50 mM Tris-HCl, pH 8.0, 500 mM KCl, 5% w/v glycerol 5 mM β-ME, 10 mM imidazole, 1 mM PMSF, and 0.1 mg ml$^{-1}$ lysozyme) by stirring at 4 °C for 60 min. The cells were lysed with one passage through a French Press high-pressure cell disrupter (Thermo Electron) at 1,500 psi; the lysate was then cleared by centrifugation at 38,000 *g* for 30 min at 4 °C. The cleared supernatant was incubated with Ni-NTA agarose beads (Qiagen) pre-equilibrated with buffer E (50 mM Tris-HCl, pH 8.0, 500 mM KCl, 5% w/v glycerol, 5 mM β-ME, 10 mM imidazole) at 4 °C for 1 h on a rotator. The beads were subsequently washed with buffer E, to which 60 mM of imidazole was added; the proteins were then eluted by a 10-min incubation with 500 mM of the imidazole-supplemented buffer E. The eluted *Msm*IMPDH samples were transferred to storage buffer (50 mM Tris-HCl, pH 8.0, 0.5 mm TCEP) using a PD10 desalting column (Cytiva), aliquoted, flash-frozen in liquid nitrogen, and stored at −80 °C. The purity of *Msm*IMPDH isolates was analyzed by SDS–PAGE; the concentration was measured using a NanoDrop spectrophotometer (Thermo Fisher).

## Expression and purification of *Mtb*IMPDH

*Mtb*IMPDH expression and purification followed the same general protocol established for our structural analysis dedicated *Msm*IMPDH. However, to improve solubility, we used the expression plasmid pET24d-HS_MtbGuaB2 encoding *Mtb*IMPDH fused with a His10x tag

flanked by *E. coli* codon-optimized SUMO protein. After the initial IMAC step, Ulp1 protease was used to remove the His10-SUMO tag.

## Enzyme kinetics

All enzymatic reactions were performed in 500 μl volumes in reaction buffer containing 50 mM HEPES (pH 7.4), 200 mM KCl, 5 mM DTT, and 15 nM *Msm*IMPDH or its mutants. When nucleotide effectors were present, the 1 mM excess of the $MgCl_2$ was used. The concentrations of the substrates and effectors are indicated in the individual experiments. The initial reaction velocity of the *Msm*IMPDH reaction was determined based on the time-dependent increase in NADH absorbance at 340 nm. The absorbance was measured in 20 s intervals for 5–15 min in a 1 cm quartz cuvette at a temperature of 37 ± 0.1 °C using a Specord 200 PLUS spectrophotometer (Analytik Jena). The initial reaction rate (in A s$^{-1}$ cm$^{-1}$) was calculated as a slope of the linear part of the steady-state reaction progress curve by least-squares linear regression. The initial reaction velocity in s$^{-1}$ units was calculated from Eq. (1):

$$v_0 = \frac{slope_E - slope_{blank}}{[E] \cdot \varepsilon} \tag{1}$$

where $slope_E$ and $slope_{blank}$ represent the slopes of the absorbance increase in A s$^{-1}$ cm$^{-1}$ units for a reaction mixture containing *Msm*IMPDH, NAD$^+$, IMP, and other tested components and a blank mixture containing only *Msm*IMPDH and the corresponding concentration of NAD$^+$ under the same conditions, respectively. [E] denotes the molar concentration of *Msm*IMPDH in nM and ε is the absorption coefficient of NADH at 340 nm in nM$^{-1}$ cm$^{-1}$ units (6.22·10$^{-6}$). The molecular weight of *Msm*IMPDH was attributed as 55,000 g mol$^{-1}$.

The substrate reaction kinetics were determined as follows: 3200 μl of a premix in the reaction buffer containing all fixed components (*Msm*IMPDH, IMP, or NAD$^+$ and effectors) was prepared. The reaction was initiated by pipetting 500 μl of the premix into 20 μl aliquots of the varied substrate at a 25-fold concentration. Six reactions were monitored spectrophotometrically in parallel using a six-position carousel. The corresponding kinetic parameters were calculated by fitting the initial velocity against the NAD$^+$ concentration plot using substrate inhibition Eq. (2) and the initial velocity against the IMP concentration plot using Hill Eq. (3):

$$v_0 = \frac{V_{max} \cdot [NAD^+]}{K_m + [NAD^+] \cdot \left(1 + \frac{[NAD^+]}{K_i}\right)} \tag{2}$$

$$v_0 = \frac{V_{max} \cdot [IMP]^{n_H}}{K_{0.5}^{n_H} + [IMP]^{n_H}} \tag{3}$$

The concentration effect of combining ATP with GTP or (pp)pGpp was tested at a fixed substrate concentration of 100 μM IMP and 2 mM NAD$^+$. The nucleotide effect on the reaction rate was expressed as the relative velocity calculated by Eq. (4):

$$relative\,v_0 = \frac{v_0^{nucleotides}}{v_0^{ctrl}} \tag{4}$$

where $v_0^{nucleotides}$ and $v_0^{ctrl}$ are initial velocities (in A s$^{-1}$) for the enzymatic reaction with and without nucleotides, respectively. All regressions were carried out using GraphPad Prism 10 software. All the error bars in the graphs and the errors in the tables represent the standard error of the mean (SE). For additional information on nucleotide handling, see the Supplementary Methods. *Mtb*IMPDH enzyme kinetics were measured using the same protocol as *Msm*IMPDH but at an enzyme concentration of 30 nM.

## IMPDH structural determination by cryo-EM

**Preparation of cryo-EM grids.** The composition of the cryo-EM buffer was identical to the enzymatic assays (50 mM HEPES, pH 7.4, 200 mM KCl, 5 mM DTT, 2 mM MgCl₂ excess over nucleotide ligands). We applied 3-μl aliquots of the *Msm*IMPDH octamer (~2.5–3.5 μM) to Quantifoil R1.2/1.3 or R2/1 Au 300 mesh grids, which were then immediately blotted for 2 s and plunged into liquid ethane using the Vitrobot Mark IV (FEI) at 4 °C and 100% humidity.

The grids were loaded into a Titan Krios electron microscope (FEI) housed at the Central European Institute of Technology, Masaryk University, Brno, Czech Republic. Operating at an accelerating voltage of 300 keV, the microscope was equipped with either a post-GIF K2 Summit or a K3 BioQuantum electron camera (Gatan) operated in counting mode. Cryo-EM data was collected using SerialEM software. For more information on data collection, see Supplementary Table 5.

**Single-particle reconstruction.** All movie frames were aligned using MotionCor2 software[38]. Thon rings obtained from the combined power spectra of every 4 e–/Å2 were employed to calculate parameters of the contrast transfer function using CTFFIND 4.1[39]. Particles were selected using Topaz (35)-trained picking models for each individual dataset. Subsequent 2D and 3D cryo-EM image processing was performed in RELION 4.0[40], as illustrated in Supplementary Figs. EM 1–14. The final cryo-EM density maps were generated by the post-processing feature in RELION and sharpened or blurred into MTZ format using CCP-EM[41]. The resolutions of the cryo-EM density maps were estimated at the 0.143 gold standard Fourier shell correlation (FSC) cutoff. The local resolution was calculated using RELION, and reference-based local amplitude scaling was performed using LocScale[42].

**Model building, refinement, and analysis.** The initial template for *Msm*IMPDH was generated using AlphaFold2[43] and rigid-body fitted into the cryo-EM maps using UCSF ChimeraX[44]. The initial structure was then improved by iterative cycles of manual fitting with Coot v.0.9.8.6[45] and ISOLDE v.1.4[46], and automated fitting with phenix.real_space_refine[47]. The relative positions of the catalytic and CBS domains were adjusted in Coot using ProSMART self-restraint in combination with whole-domain morphing. Where the position of the flexible loops differed from the prediction (residues 400–450), it was automatically built using Buccaneer software within the CCP-EM software suite[48] and manually adjusted in Coot for the best fit. All water molecules around the ligand binding sites and active sites were manually placed in Coot. Mg²⁺ cations were added according to published homologous structures and validated according to coordination geometry and coordination distances. Molecular restraints for nucleotide ligands were generated using the Grade2 tool on the Grade Web Server (Global Phasing). Model quality and model-versus-data fit statistics were validated using the Phenix comprehensive Cryo-EM validation tool.

Data collection parameters and refinement statistics are summarized in Supplementary Table 5.

Structural figures were generated in PyMOL v.2.5.5 (Schrödinger). The two-dimensional diagram of active site coordination was created using PoseView[49].

## Hydrogen/deuterium exchange

For HDX labelling experiments, *Msm*IMPDH at a concentration of 40 μM was used individually and in ligand mixtures of the following compositions: equilibration solution in ATP-binding experiment: 40 μM *Msm*IMPDH, 20 mM ATP and 22 mM Mg²⁺; equilibration solution in ATP + GTP-binding experiment: 40 μM *Msm*IMPDH, 5 mM ATP, 20 mM GTP, 10 mM IMP, and 27 mM Mg²⁺. The compositions were calculated using a conservative approximate K_D value of 1,000 μM to ensure at least 80% of *Msm*IMPDH remained in the complex formed

with the applied ligands. The samples were mixed with D₂O-based buffer (50 mM Tris-HCl, pH 8.0) in a 1:4 ratio, incubated at 4 °C for 0, 2, 5, 10, 20, or 120 s, and then quenched with the addition of an equal volume of quench buffer (0.1 mM phosphate, pH 2.51). Samples were then frozen in liquid nitrogen and stored at −80 °C.

## LC–MS analysis

Peptides were identified by tandem mass spectrometry of non-deuterated *Msm*IMPDH. Samples were injected into a refrigerated NanoAcquity UPLC M-Class System (Waters) with chromatographic elements held at 0 °C. The samples were then passed through a pepsin/nepenthesin-2 protease column (Affipro) for 3 min at 15 °C; the generated peptides were trapped and desalted on an Acquity UPLC BEH C18 VanGuard pre-column (Waters). Peptides were then separated in a gradient of acetonitrile for 12 min. MS spectra were acquired with a Synapt G2-Si mass spectrometer equipped with electrospray ionization (ESI) and quadrupole/time-of-flight (Q/TOF) (Waters); data were processed on the ProteinLynx Global Server (PLGS) v.3.0.2 (Waters). DynamX 3.0 (Waters) was used to filter peptides and then to determine HDX differences between the examined protein states. The LC analysis of labelled samples was identical to that of non-deuterated samples. The HDX experiments were performed in triplicate for each labelling time point. Additional information on the LC–MS analysis, configurations, and parameters used for PLGS and DynamX software, as well as the statistical evaluation of data, can be found in the Supplementary Methods and in the Supplementary Table 9.

## DRaCALA assay

Differential radial capillary action of ligand assay (DRaCALA)[50] is a technique used to separate the free ligand from bound protein–ligand complexes on a dry nitrocellulose membrane. Our procedure involved the preparation of a 10-ml mixture containing 50 μM *Msm*IMPDH, 6 nM radiolabelled ATP [γ-33P] (0.4 mCi; Hartmann Analytic) in 50 mM HEPES (pH 7.5), 200 mM KCl, 1 mM MgCl₂, and 0.5 mM TCEP. Additionally, mixtures containing 1 mM Mg:ATP or Mg:GTP were prepared in parallel. The samples were incubated for 10 min at room temperature, with 2.5 μl aliquots spotted on the nitrocellulose membrane. After 30 min of drying, the membrane was exposed using a phosphor screen cassette (Amersham Biosciences). The signal was then developed using the Typhoon scanner (Amersham Biosciences).

## Small angle X-ray scattering

All SAXS measurements were performed on the SAXSpoint 2.0 (Anton Paar) system equipped with the MetalJet C2 X-ray source (Exillum) and the Eiger R 1 M detector (Dectris) at the Centre of Molecular Structure (Institute of Biotechnology of the Czech Academy of Sciences, Vestec, Czech Republic). Measurements incorporated a sample-to-detector distance of 0.8 m and an X-ray wavelength of λ = 0.134 nm. Samples of 3 mg ml⁻¹ *Msm*IMPDH in buffer containing 50 mM HEPES, pH 7.4, 200 mM KCl, and 0.5 mM TCEP were measured in the presence of ATP, GTP, and IMP (0–10 mM) and a 2 mM molar excess of MgCl₂ over the nucleotide concentration. Buffer-solution datasets were collected under the same conditions, including the respective nucleotides. Data background subtraction and analysis were performed using the PRIMUS program as part of the ATSAS software suite[51]. The radius of gyration was calculated using Guinier approximation, with pair-distance distribution function and D_{max} parameters analyzed using the GNOM program[52]. The ratio of compressed/extended fractions in the ATP + IMP sample was calculated using the OLIGOMER program[53]. Theoretical SAXS profiles were calculated from corresponding cryo-EM structures and fitted to the experimental curves using the CRYSOL program[54]. Dummy-atom models were calculated ab initio by executing 32 runs on the DAMMIF program[55], which were then compared and averaged using DAMAVER. The resulting model was used as the starting model for the final run using DAMMIN[56]. SAXS data collection

and processing parameters are summarized in Supplementary Table 10 and Supplementary Figs. SAXS 1–5.

## Mass photometry analysis

All mass photometry experiments were performed using the TwoMP automated mass photometer (Refeyn) at room temperature. The instrument was calibrated following the manufacturer's protocols using bovine serum albumin (BSA) (monomers: 66 kDa; dimers: 132 kDa) and IgG (monomers: 150 kDa; dimers: 300 kDa). *Msm*IMPDH protein samples were diluted with buffer containing 50 mM HEPES, pH 7.4, 200 mM KCl, and 0.5 mM TCEP at a final protein concentration of 20 nM. The effects of nucleotides on oligomeric status were assessed in the presence or absence of ATP, GTP, and IMP (5 mM) and a 2 mM molar excess of $MgCl_2$ over the nucleotide concentration, following a 5 min incubation. Scattering movies were recorded for 60 s and analyzed using DiscoverMP software (Refeyn).

## IMPDH phylogenetic analysis

Annotated genomic data, translated into protein sequences, were obtained using ENTREZ API[57] for all available representatives of the Actinomycetia class. Unclassified organisms were filtered out from the genomes obtained. Additionally, we retained only one representative for each relative species denoted by the abbreviation "sp". The final filter applied to the remaining genomes involved checking for the presence of similar sequences to GuaB2 from *M. smegmatis* MC² 155 (A0QSU3) using the BLAST algorithm[58] implemented using the Biopython bioinformatics library[59]. Assuming that representatives of actinomycetes have three similar GuaB-family sequences (GuaB1, GuaB2, and GuaB3), the genomes selected for further processing needed to contain at least three hits, likely indicating the presence of these sequences. Of the original 6480 genomes, 2846 genomes were retained for final processing. Of these, the GuaB2 sequence was identified using the PSI-BLAST[60] algorithm based on HMM profiling implemented in Biopython using the wrapper for NCBI BLAST+ tools[61]. Multiple sequence alignment of the GuaB2 sequences was performed using the MAFFT algorithm v.7.407[62] (default settings for gap extension penalty and gap opening penalty, and the BLOSUM matrix were applied), again using the command line wrapper for Biopython.

We used the Python version of the seqLogo R library v.1.63.0[63] for graphical representation of amino acid conservation. Information content is measured in bits. Amino acid residues with an equal probability of occurring at a given position have an information value of zero bits. Conversely, a position occupied by only one amino acid has an information value of four bits. The information content at position $w$ in the displayed motif is given by the formula:

$$IC(w) = \log_2 J + \sum_{j=1}^{J} p_{wj} \log_2 p_{wj} = \log_2 J - entropy(w) \quad (5)$$

where $J$ denotes the number of characters, $w$ the position in the sequence, and $j$ the index of the character in the character's alphabet.

## Reporting summary

Further information on research design is available in the Nature Portfolio Reporting Summary linked to this article.

## Data availability

The atomic coordinates are deposited in the Research Collaboratory for Structural Bioinformatics Protein Data Bank (RCSB PDB) with accession codes: PDB 8PW3 (*Msm*GuaB2 apo), PDB 8Q65 (*Msm*GuaB2-ATP), PDB 8QQV (*Msm*GuaB2-ATP + IMP extended), PDB 8QQW (*Msm*GuaB2-ATP + IMP compressed), PDB 8QQX (*Msm*GuaB2-ATP + IMP intermediate), PDB 8QQP (*Msm*GuaB2-ATP + GTP compressed), PDB 8QQQ (*Msm*GuaB2-ATP + GTP less compressed), PDB 8QQR (*Msm*GuaB2-ATP+ppGpp compressed), PDB 8QQT (*Msm*GuaB2 -ATP +ppGpp less compressed). The cryo-EM maps are deposited in the Electron Microscopy Data bank (EMDB) under accession codes: EMD–17988 (*Msm*GuaB2 apo), EMD–18184 (*Msm*GuaB2-ATP), EMD–18606 (*Msm*GuaB2-ATP + IMP extended), EMD–18607 (*Msm*GuaB2-ATP + IMP compressed), EMD–18608 (*Msm*GuaB2-ATP + IMP intermediate), EMD–18600 (*Msm*GuaB2-ATP + GTP compressed), EMD–18601 (*Msm*GuaB2-ATP + GTP less compressed), EMD–18602 (*Msm*GuaB2-ATP+ppGpp compressed), EMD–18604 (*Msm*GuaB2 -ATP+ppGpp less compressed). Data on the SAXS experiments are deposited in the Small Angle Scattering Biological Data Bank (SASBDB) under the following accession codes: SASDUM5 (*Msm*GuaB2 apo), SASDUN5 (*Msm*GuaB2 IMP), SASDUP5 (*Msm*GuaB2 ATP), SASDUQ5 (*Msm*GuaB2 ATP + IMP) and SASDUR5 (*Msm*GuaB2 GTP). The source data for all biochemical assays have been deposited in the Zenodo repository and are available at the following URL: https://doi.org/10.5281/zenodo.12646467. The HDX-MS datasets generated in this study have been deposited in the Zenodo repository and are available at the following URL: https://doi.org/10.5281/zenodo.12697518.

## Code availability

The code for *Msm*IMPDH phylogenetic data analysis that supports this study is available on the GitHub platform: https://doi.org/10.5281/zenodo.11047778.

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

## Acknowledgements
This work was supported by the National Institute of Virology and Bacteriology (EXCELES, LX22NPO5103), funded by the EU's Next Generation EU programme. We gratefully acknowledge the assistance of the Czech Infrastructure for Integrative Structural Biology (CIISB), the Czech arm of the European Research Infrastructure Consortium's Integrated Structural Biology Infrastructure (Instruct-ERIC), funded by the Ministry of Education, Youth and Sports of the Czech Republic (LM2023042) and the European Regional Development Fund (UP CIISB, CZ.02.1.01/0.0/0.0/18_046/0015974). This support enabled us to perform measurements at the Cryo-Electron Microscopy and Tomography Core Facility at the Central European Institute of Technology at Masaryk University (CEITEC MU) and at the Biophysical Techniques and Diffraction Techniques Core Facilities at the Centre of Molecular Structure at the Biotechnology and Biomedicine Centre in Vestec (BIOCEV). We are especially grateful to the following scientists at these core facilities: T. Charnavets (mass photometry), J. Stránský (SAXS measurements), and J. Nováček (cryo-EM experiments). We acknowledge the Electron Microscopy Core Facility at the Institute of Molecular Genetics (IMG) in Prague, funded by MEYS (LM2023050) and ERDF (CZ.02.1.01/0.0/0.0/18_046/0016045, CZ.02.01.01/00/23_015/0008205) for their support with obtaining the electron microscopy data presented herein. We thank D. Galiana for her technical support and M. Hubálek for his advice on the MS analysis. The authors are grateful to M. Doležal for his insights on structural determination and to V. Veverka for his valuable feedback on the manuscript. O.B. acknowledges a PhD fellowship from the University of Chemistry and Technology, Prague.

## Author contributions
O.B., T.K., and A.F. determined the cryo-EM structures. O.B. and M.Č. purified the proteins. Z.K. developed and performed the in vitro biochemical assays and DRaCALA. O.B. performed the SAXS and MP experiments. J.S. conducted the HDX–MS analysis. K.C. provided and analyzed the phylogenetic data. D.R. provided the nucleotides, T.K. supervised the structural studies, and Z.K. and I.P. developed the concept for the study. O.B. and Z.K. designed the experiments and performed the data analysis and interpretation. O.B., T.K., Z.K., and I.P. wrote the manuscript. All authors have read and commented on the manuscript.

## Competing interests
The authors declare no competing interests.
