## [Peer Review File · Nature Communications]

Deciphering the allosteric regulation of mycobacterial inosine-5'-monophosphate dehydrogenaseREVIEWER COMMENTS

Reviewer #1 (Remarks to the Author):

In this manuscript, Bulvas et al. have characterized the regulation of *Mycobacterium smegmatis* IMPDH (MsmIMPDH) by different nucleotides, GTP and (p)ppGpp in the absence or in the presence of ATP. The authors have performed enzyme kinetics experiments and solved by cryo-EM the 3D-structures of several complexes of MsmIMPDH with different combinations of IMP, ATP, GTP and ppGpp. They claim that their results are novel and could be expanded to other bacterial IMPDHs. However, their results are only valid for the MsmIMPDH. Moreover, as detailed below, I have many concerns about this manuscript, and the overall advance of the paper is not sufficient for publication in *Nature Communications*. The paper should be submitted to a more specialized journal.

Firstly, the authors do not take into account all the data available in the literature. Some examples are listed below:

- in the summary, the following sentences "the precise molecular mechanism of IMPDH regulation in bacteria remains unclear" and "... new possibilities for the development of allosteric inhibitors with antibacterial potential" need modification. Several papers by different teams have described the regulation of bacterial IMPDHs at the molecular level, and different chemical series have been identified acting as allosteric inhibitors.
- in the introduction, the authors cite the paper of Singh et al. which considers GubB from *M. tuberculosis* as a vulnerable target. However, they omit the work by Park et al. (2017 *ACS Infectious Diseases* DOI: 10.1021/acsinfecdis.6b00103), which reaches an opposite conclusion, i.e., IMPDH is not validated as a target for the development of antituberculosis drugs.
- in the introduction, the classification of bacterial IMPDHs (based on kinetics and oligomeric states) into class I and class II reported in ref 6 should be included and detailed, before to introduce the divergence for the (p)ppGpp binding site.
- at the end of the first paragraph of the results section, the authors claim that their findings "contrast with other reported prokaryotic IMPDHs". However, Fernandez-Justel et al. (ref 9) have also shown that GTP and (p)ppGpp both inhibit *Streptomyces coelicolor* IMPDH catalytic activity in the presence of ATP.
- the authors also do not comment on the cooperative behavior of MsmIMPDH. The *M. tuberculosis* homologue has been reported by Rostirella et al. (ref 10) to exhibit a sigmoidal curve for the velocity data as a function of IMP concentration. On the other hand, Usha et al. (ref 11) as well as Chen et al. (ref 26) have clearly shown a Michaelis-Menten plot for the velocity data for both substrates. Linked to this, the authors have not explained the rationale for choosing "the model bacterial species *M. smegmatis* IMPDH" instead of the *M. tuberculosis* one. These points need clarification.
- concerning the structural data, at least two papers (ref 2 and Anthony et al. 2017 DOI: 10.1091/mbc.E17-04-0263) and one review (Burrell and Kollman 2022 DOI: 10.1042/bst20210446) have to be taken into account as they detail the regulation of the quaternary structures of the two human IMPDHs by nucleotides (IMP, GTP, ATP) at the molecular level. Comparative analysis needs to be included in the figures, which generally require improvement. Additionally, supplemental panels should be added to strengthen the manuscript.
- the catalytic mechanism of the IMPDH reaction has been characterized at the molecular level (Josephine et al. DOI: 10.1021/bi101590c). In the second paragraph of "IMP induces MsmIMPDH octamer extension", the authors should compare their data to that provided by Hedstrom's team.
- an in-depth phylogenetic analysis has been published recently by Buey et al. (ref 2; Figure 5). What is the novelty of Sup Figure 1?

Second, the authors have not performed any biophysical investigation to support their conclusions regarding the observed oligomeric states. Rostirella et al. (ref 10) have reported the presence of different oligomeric states for the mycobacterial IMPDH, particularly tetramers. Performing analytical ultracentrifugation experiments, at the very least, would be a means to corroborate the quaternary structures observed by cryo-EM. Additionally, concerning the fluctuation of overall structural rearrangements, SAXS data could confirm the existence of extended or compressed conformations in solution, depending on the added nucleotide.

Third, based on their structural data, the authors have mutagenized some residues. However, for all these MSM IMPDH mutants, no kinetic parameters nor any data on their oligomeric states have been given in the manuscript. Some of these mutants such as R157A, D158A, E200A and K201A are supposed to be impacted in their ATP binding: how did the authors check the ATP binding?

Concerning the 1 mM MgCl₂ excess, it has been reported by Evrin et al. (2007 DOI: 10.1074/jbc.M606963200) that "an acceptable compromise was to use a 2 mM excess of MgCl₂ above the concentration of NTPs". A 2 mM MgCl₂ excess should be tested. For the GTP or ppGpp inhibitory studies (Sup Figures 9 and 12), why was an IMP concentration (2 mM) chosen to be 14 times higher than the measured K_{0.5} and an NAD concentration (100 μM) only 0.1 that of the K_m value?

Minor points:

- throughout the manuscript, several spelling, grammatical, and stylistic errors need to be corrected. They are also weird notations for the enzymatic part (V_{lim} or K_h), the conventional notations (V_{max} and K_{0.5}) should be used. Check the notation for the phosphate buffers: what is KPO₄? Tris pH8 with HCl (if yes, then Tris-HCl)?
- In the "IMPDH phylogenetic analysis", correct the GuaB (and not Guab or GuB); in this same §, if genes are taken into consideration instead of proteins, then the notation should be *guaB* (italics and g in lower case). Again, in this §, last sentence, what "j the index of the character in the character's alphabet" means?
- Figure 5 should be modified so that all the residues listed in the text are also represented in the different panels. The angle values in panel d should be added.
- Figure 6 is confusing. It seems that IMPDH is tetrameric and that in the presence of IMP the tetramer dissociates into dimers.

Reviewer #2 (Remarks to the Author):

This manuscript describes mechanisms of allosteric regulation of mycobacterial IMPDH by adenine and guanine nucleotides - a well-studied phenomenon in other species, but here there is a clear description of the mechanism by which ATP/GTP ratios are sensed by the enzyme that is novel and should be of specific interest to the field. It also establishes a role for ppGpp in regulating mycobacterial IMPDH. More broadly, the description of complex structural regulatory mechanisms is likely to be of interest to a wider audience interested in principles of allosteric regulation.

The structural studies by cryo-EM are well done, and support the conclusions. Functional enzyme activity data and mutational analysis are consistent with the structural data, and HDX is used to confirm relative stability of regions that is consistent with the cryo-EM structures.

The paper is well written and clear. I have only a few minor comments that I think may improve the analysis:

1) The findings will be of interest to IMPDH aficionados generally, and are consistent with multiple structural studies of eukaryotic IMPDH in recent years. The discussion would benefit significantly from a comparison of the structural rearrangements observed here with those seen in eukaryotic IMPDHs, specifically:

- how do the compressed GTP/ppGpp bound structures compare with eukaryotic compressed structures? Does the compressed or "super-compressed" conformation more closely resemble those?
- variability if the conformational state of ATP/IMP-bound human IMPDH was reported several years ago (Johnson & Kollman, eLife 2020). How do the ATP/IMP-bound mycobacterial structures compare to those in terms of extension/compression of the interdomain hinges?
- the descriptions of ligand binding (IMP/ATP/GTP/ppGpp) are very clear, but it would be useful to know how these compare to prior structures with these ligands bound (one assumes they are identical, but if not this would be very interesting)

- are the rearrangements of finger, flap, and Cys-loops conserved across these structures?

2) It would be interesting to discuss whether there are conformational changes in the arrangement of the tetrameric catalytic domains that accompany IMP binding and/or ATP/GTP binding. Two conformations were reported in the Johnson & Kollman paper (and subsequent papers) associated with different activity levels, described as "bowed" and "flat" and I'm wondering if a similar transition occurs in the mycobacterial structures.

3) In figure 4 it would be helpful to specify for each panel exactly which structures are being compared - I was unsure, for example, in 4a whether these were two different classes from the ATP/IMP dataset, or this was ATP vs ATP/IMP?

4) The authors present new data (Suppl. Fig. 14) in the discussion which is not discussed in results. I understand that the results are somewhat tangential to their overall story, but these should be introduced somewhere in the results section (or omitted - but I think this will be useful information for the field).

Reviewer #3 (Remarks to the Author):

The manuscript by Bulvas et al describes the allosteric regulation of mycobacterial IMPDH2. Although this enzyme has received attention as a promising target for new antimycobacterial agents, the effects of ATP, GTP and ppGpp on enzymatic activity have not been previously investigated. Thus this work is a very important contribution. The manuscript reports a thorough characterization of allosteric regulation, including enzyme kinetics, H/D exchange and cryoEM, with the unanticipated finding of ATP-dependent GTP inhibition that is further enhanced by ppGpp. The cryoEM and H/D exchange experiments provide the structural context for these findings. The experiments are carefully performed and the paper is very well written. Below are a couple questions/comments that should be addressed.

Major points

1. The values for K_m , n_H etc should be included in the text.

2. Hill kinetics for IMP dependence have not generally been observed in other IMPDHs. The authors are careful to include constant excess Mg relative to the ATP/GTP/ppGpp concentrations. However, IMP and NAD can also chelate Mg. Could Mg be the cause of the cooperativity? Is cooperativity observed in the absence of Mg?

3. The authors would like to argue that GTP inhibition is physiologically relevant. They previously reported the intracellular concentrations of ATP and GTP were 4 mM and 2 mM, respectively, in Knejzlik et al 2021. Therefore it would be nice if Fig.1b and d included GTP = 2 mM.

4. The inactive open conformation of the Cys loop appears to be incompatible with K^+ binding. Was there K^+ in the cryoEM experiments? Please include the buffer for these experiments in the Methods.

5. line 434-435: Contrary to the author's claim, obtaining selectivity has not been a challenge for inhibitors that bind to the active site of mycobacterial IMPDHs- in fact, both of the papers cited describe how the cofactor sites of human and bacterial IMPDHs are highly diverged. Selective inhibitors that bind in the cofactor site have been reported- for example PMID: 29746130 and PMID: 26440283. That none of these have entered the clinic is due to the many other challenges of drug discovery- which is reason enough to need new approaches.

6. It would be nice if the authors included a comparison of their findings with the allosteric regulation of the human enzymes.

Minor points

1. The experiments with varied IMP were performed at fixed NAD = 2 mM, which is approximately the value of K_m . Since the fixed NAD concentration was not saturating, the experiments do not really measure "kcat".
2. It would be nice if the plots showed the concentration of fixed reagents, for example ATP and NAD in e.g., Fig.1a (as the authors do in the Supplementary figures).
3. line 213: "a notable increase in deuterium uptake" relative to what?
4. Fig 3d. What are RFU?
5. Fig. 4e: does the distance line simply show 10Å, or does it mean those two helices have moved 10Å? If the former, it might be better to have the line on the far right of the figure rather than after the arrow and remove the "delta".
6. Line 331: the authors should describe what mutants were characterized
7. Supplementary Table 2, 2nd GTP entry, MgCl₂ should presumably be "2" not "1".
8. The term "Kh" is used throughout the manuscript but K_{0.5} is used in the equation.

Responses to reviewer comments:

Reviewer #1 (Remarks to the Author):

In this manuscript, Bulvas et al. have characterized the regulation of Mycobacterium smegmatis IMPDH (MsmIMPDH) by different nucleotides, GTP and (p)ppGpp in the absence or in the presence of ATP. The authors have performed enzyme kinetics experiments and solved by cryo-EM the 3D-structures of several complexes of MsmIMPDH with different combinations of IMP, ATP, GTP and ppGpp. They claim that their results are novel and could be expanded to other bacterial IMPDHs. However, their results are only valid for the MsmIMPDH. Moreover, as detailed below, I have many concerns about this manuscript, and the overall advance of the paper is not sufficient for publication in Nature Communications. The paper should be submitted to a more specialized journal.

Response: We thank the reviewer for the critical comments on our paper and for suggesting experiments that would support our results. Based on the reviewer's recommendations, we performed a further biophysical and biochemical characterization of *MsmIMPDH*. The structural rearrangements in the presence of ATP and IMP were analyzed using the SAXS method, and the oligomeric state was profiled using mass photometry. We confirmed the binding properties for ATP of *MsmIMPDH* Site 1 mutants using the DRaCALA method. Furthermore, we isolated and kinetically characterized IMPDH from *M. tuberculosis*. The details of these experiments are described in more detail in the text below. Additionally, we compared the mechanism of allosteric regulation of *MsmIMPDH* with that of human IMPDH in the discussion.

We respectfully disagree with the comment about the limited validity of our findings. Our original manuscript has already demonstrated the conservation of ligand binding sites in actinobacterial IMPDHs (Supplementary Fig. 13 of the previous version, Supplementary Fig. 19 of the revised version). This suggests that other actinobacterial species employ a similar binding and allosteric regulatory mechanism using ATP, GTP, and ppGpp. Our revised version now includes the kinetic characterization of IMPDH from *M. tuberculosis* (Supplementary Figs. 8 and 9 and Supplementary Table 4), demonstrating very similar catalytic behaviour, IMP cooperativity, and response to ATP, GTP, and ppGpp when compared with *MsmIMPDH*.

The main novelty of our work lies in the comprehensive analysis of the allosteric regulation mechanism of *MsmIMPDH*. Our extensive structural data on *MsmIMPDH* complexes (with different ligands), detailed kinetic analysis, and biophysical characterization demonstrate the following: *MsmIMPDH* is regulated by both GTP and ppGpp in the presence of ATP; IMP binding (not ATP) plays an important role in the extension of *MsmIMPDH* octamers; and the basic residues in the hinge regions involved in GTP and ppGpp binding are key to this regulatory mechanism.

Firstly, the authors do not take into account all the data available in the literature. Some examples are listed below:

- in the summary, the following sentences “the precise molecular mechanism of IMPDH regulation in bacteria remains unclear” and “... new possibilities for the development of allosteric inhibitors with antibacterial potential” need modification. Several papers by different teams have described the regulation of bacterial IMPDHs at the molecular level, and different chemical series have been identified acting as allosteric inhibitors.

Response: The state of knowledge about IMPDH regulation is summarized in the Introduction section, where the aforementioned papers on bacterial IMPDH regulation are referenced. To the best of our knowledge, the exact molecular mechanism for how the nucleotides bound in the CBS domain cause the inhibition and what drives the IMPDH octameric compression/extension remain rather unclear.

Based on our search of the literature, the only allosteric inhibitors of bacterial IMPDHs have been published by Alexandre et al. (Ref. 36 in the Discussion section). These inhibitors specifically target only the allosteric site 1 of class I IMPDHs and thus block the activation by the natural positive effector ATP. To date, no allosteric inhibitors targeting canonical Site 2 have been published.

- in the introduction, the authors cite the paper of Singh et al. which considers GuaB3 from M. tuberculosis as a vulnerable target. However, they omit the work by Park et al. (2017 ACS Infectious Diseases DOI: 10.1021/acsinfecdis.6b00103), which reaches an opposite conclusion, i.e., IMPDH is not validated as a target for the development of antituberculosis drugs.

Response: We thank the reviewer for the comment. We have added a comment on the contradictory work by Park et al. to the Introduction. The additional reference to Hedstrom et al. (2018 J. Med. Chem; DOI: 10.1021/acs.jmedchem.7b01839) has been provided.

Revised sentence: “Mycobacterial IMPDH is considered by some authors to be a promising target for the treatment of infections caused by pathogenic *Mycobacterium* species, although others have raised concerns about the efficacy of such a potential treatment.”

- in the introduction, the classification of bacterial IMPDHs (based on kinetics and oligomeric states) into class I and class II reported in ref 6 should be included and detailed, before to introduce the divergence for the (p)ppGpp binding site.

Response: The mentioned classification based on Alexandre et al. (now Ref 8) has been added to the Introduction section.

Added sentence: “Alexandre et al.⁸ propose a classification of bacterial IMPDHs, dividing them into two classes. Class I IMPDHs require ATP to bind to the CBS domain for full activity and

always exist in an octameric form. In contrast, class II enzymes, which occur as tetramers, do not require ligand binding for activation and undergo ATP-induced octamerization.”

- at the end of the first paragraph of the results section, the authors claim that their findings “contrast with other reported prokaryotic IMPDHs”. However, Fernandez-Justel et al. (ref 9) have also shown that GTP and (p)ppGpp both inhibit *Streptomyces coelicolor* IMPDH catalytic activity in the presence of ATP.

Response: We respectfully disagree with this comment regarding the work of Fernandez-Justel et al. (now Ref 11). In the respective work, they conclude that GTP is only likely to have a weak inhibitory effect on enzymes in *Bacillus subtilis* (Firmicutes) and *Streptomyces coelicolor* (Actinobacteria). Indeed, in their follow-up review on the topic (Ref 2), they state that “in contrast to Proteobacteria, in most other bacterial phyla, including Firmicutes and Actinobacteria, GTP does not bind to the second canonical site and, therefore, does not affect IMPDH activity”. The idea that GTP should only inhibit proteobacterial (and not actinobacterial) IMPDHs is summarized in Ref. 2; Fig. 5. Therefore, we believe our results contradict the currently accepted hypothesis.

- the authors also do not comment on the cooperative behavior of *Msm*IMPDH. The *M. tuberculosis* homologue has been reported by Rostirella et al. (ref 10) to exhibit a sigmoidal curve for the velocity data as a function of IMP concentration. On the other hand, Usha et al. (ref 11) as well as Chen et al. (ref 26) have clearly shown a Michaelis-Menten plot for the velocity data for both substrates. Linked to this, the authors have not explained the rationale for choosing “the model bacterial species *M. smegmatis* IMPDH” instead of the *M. tuberculosis* one. These points need clarification.

Response: We thank the reviewer for raising this important point.

To address the question of the compatibility of *Msm*IMPDH model with that of *M. tuberculosis* IMPDH (*Mtb*IMPDH), we isolated *Mtb*IMPDH and performed a basic biochemical characterization. Our experiments underline the very similar catalytic behaviour of *Mtb*IMPDH and *Msm*IMPDH, including the IMP cooperative behaviour and inhibition by purine nucleotides. The data have been added to the Results section as well as Supplementary Figs. 8 and 9 and Supplementary Table 4. The additional experiments with *Mtb*IMPDH show that the knowledge gained from *Msm*IMPDH should be directly applicable to *Mtb*IMPDH. The use of *M. smegmatis* as a model *Mycobacterium* is well-established (see, for example, Sparks et al. 2023, doi: 10.1128/jb.00337-22).

Regarding the cooperativity of *Mtb*IMPDH documented in Rostirella et al., Usha et al. and Chen et al. (now Refs 12, 13 and 34, respectively), we can only speculate that the discrepancies are caused by the reaction conditions used for the respective enzymatic assays. However, our experimental results align with those of Rostirella et al. (Ref. 12), demonstrating remarkably similar kinetic parameters.

- concerning the structural data, at least two papers (ref 2 and Anthony et al. 2017 DOI: 10.1091/mbc.E17-04-0263) and one review (Burrell and Kollman 2022 DOI: 10.1042/bst20210446) have to be taken into account as they detail the regulation of the quaternary structures of the two human IMPDHs by nucleotides (IMP, GTP, ATP) at the molecular level. Comparative analysis needs to be included in the figures, which generally require improvement. Additionally, supplemental panels should be added to strengthen the manuscript.

Response: We agree with the reviewer that the paper would benefit from a comparison with the regulation of eukaryotic IMPDHs. The corresponding comparative analysis has been added to the Discussion section (lines 456-479) as well as Supplementary Fig. 17 and 18.

Though the general mechanism behind IMPDH regulation seems to be conserved from bacteria to eukaryotes, the molecular details differ. Two interesting points emerged from our comparison:

- Two IMPDH conformations described as “bowed” and “flat” have been documented in the case of filamenting IMPDHs. While all of our MsmIMPDH structures are not directly comparable to the flat or bowed conformations seen in eukaryotic IMPDHs, they also show some degree of flexibility within the tetramer, reminiscent of the flat and bowed conformational transitions (Supplementary Fig. 17). However, the relative range of those movements is comparatively limited.
- We compared the hinge lock mechanisms of eukaryotic and mycobacterial IMPDHs (Supplementary Fig. 18). While the binding sites of GTP and ppGpp differ with respect to the IMPDHs, the general mechanism behind the locking of the hinges is distinctly similar.

- the catalytic mechanism of the IMPDH reaction has been characterized at the molecular level (Josephine et al. DOI: 10.1021/bi101590c). In the second paragraph of “IMP induces MsmIMPDH octamer extension”, the authors should compare their data to that provided by Hedstrom’s team.

Response: We have added a comparison with the results of Josephine et al. (Ref 17) and several other published substrate-bound IMPDH structures to the Results section (lines 284-286).

Added sentence: “The binding mode of IMP to MsmIMPDH is consistent with previously published structures of both prokaryotic and eukaryotic IMPDHs^{3,17-20}.”

The conformation of the active site in our structure with bound IMP is almost identical to the pre-reaction substrate-bound state of TfoIMPDH reported in the work by Josephine et al. However, since all the reported TfoIMPDH structures are in the tetrameric state and lack CBS domains, we could not compare the tetramer–tetramer interface. The main novelty of our results is the effect of the active site’s IMP-induced reorganization on the overall octamer conformation.

- an in-depth phylogenetic analysis has been published recently by Buey et al. (ref 2; Figure 5). What is the novelty of Sup Figure 1?

Response: The purpose of Supplementary Fig. 1 is not to present entirely new data, but rather to provide context for the experiments that follow and to frame the alignments from a different angle. The similar alignments were indeed presented in Buey et al. (Ref 2), but the authors did not comment on the apparent conservation of canonical Site 2 within bacteria, including Actinobacteria.

Second, the authors have not performed any biophysical investigation to support their conclusions regarding the observed oligomeric states. Rostirella et al. (ref 10) have reported the presence of different oligomeric states for the mycobacterial IMPDH, particularly tetramers. Performing analytical ultracentrifugation experiments, at the very least, would be a means to corroborate the quaternary structures observed by cryo-EM. Additionally, concerning the fluctuation of overall structural rearrangements, SAXS data could confirm the existence of extended or compressed conformations in solution, depending on the added nucleotide.

Response: We agree with the reviewer that the conclusions of our manuscript would be strengthened by the addition of complementary biophysical methods. First, we have added the analysis of *Msm*IMPDH oligomeric states in response to the nucleotides using mass photometry (Fig. 4g; lines 308-312 in the Results section). Secondly, the conformational transitions of *Msm*IMPDH compressed and extended octamers were verified using SAXS (Fig. 4f; lines 303-308 in results). The results of those additional experiments align well with our cryo-EM data. Given that *Msm*IMPDH is fully octameric in the presence of ATP (always present in the cell), we argue that these octamers reflect the major biologically relevant oligomeric state of the enzyme.

Third, based on their structural data, the authors have mutagenized some residues. However, for all these MSM IMPDH mutants, no kinetic parameters nor any data on their oligomeric states have been given in the manuscript. Some of these mutants such as R157A, D158A, E200A and K201A are supposed to be impacted in their ATP binding: how did the authors check the ATP binding?

Response: We have now added the kinetic parameters for all the mutated residues (Supplementary Table 6). In addition, the impaired binding of ATP to the above mutated variants was verified using DRaCALA (Supplementary Fig. 13a). The oligomeric state of these mutated variants was verified by mass photometry (Supplementary Fig. 13b). The mutated variants do not respond significantly to ATP-induced octamerization.

Concerning the 1 mM MgCl₂ excess, it has been reported by Evrin et al. (2007 DOI: 10.1074/jbc.M606963200) that “an acceptable compromise was to use a 2 mM excess of MgCl₂ above the concentration of NTPs”. A 2 mM MgCl₂ excess should be tested.

Response: We thank the reviewer for the insightful comment. We agree with the reviewer's assessment of the importance of Mg^{2+} ions, which are the most abundant divalent cations *in vivo*. The concentration of free Mg^{2+} in the cytoplasm of bacteria varies between 1 and 2 mM, depending on the cultivation conditions and genetic background (Alatossava et al., 1985; Froschauer et al., 2004; Szatmári et al., 2020). ATP, like other (d)NTPs, acts as a relatively strong Mg^{2+} chelator. The dissociation constant (K_d) of the Mg–ATP complex *in vivo* has previously been set at 35 μ M (Gout et al., 2014). Therefore, we chose a lower magnesium excess value (1 mM) within the reported free *in vivo* concentration range.

Nevertheless, in order to gain further insights into the effect of Mg^{2+} ions on *Msm*IMPDH activity and the effect of excess Mg^{2+} ions on ATP/GTP inhibition, we performed a series of experiments. The results are presented in Figs. Rev1_1 and Rev1_2 (see below). These data indicate that the sole ATP/GTP combination (without Mg^{2+} , represented by the blue symbol/line) has only a partial inhibitory effect on IMPDH activity or IMP kinetics. However, this effect is more pronounced in the presence of Mg^{2+} . Moreover, our data show that an excess of 1 mM Mg^{2+} relative to NTPs has an effect comparable to that of 2 mM (orange vs. red symbol/line) and is more efficient than the equimolar Mg:NTP ratio (green symbol/line). Note that these figures are not included in the current version of the manuscript. Our only intention was to fully address the comment raised by the reviewer.

Alatossava T, Jütte H, Kuhn A, Kellenberger E. **Manipulation of Intracellular Magnesium Content in Polymyxin B Nonapeptide-Sensitized *Escherichia Coli* by Ionophore A23187.** *Journal of Bacteriology* 162, 1985: 413–19. <https://doi.org/10.1128/jb.162.1.413-419.1985>.

Froschauer, EM, Kolisek M, Dieterich F, Schweigel M, Schweyen RJ. **Fluorescence Measurements of Free [Mg²⁺] by Use of Mag-Fura 2 in *Salmonella Enterica*.** *FEMS Microbiology Letters* 237, 2004: 49–55. <https://doi.org/10.1111/j.1574-6968.2004.tb09677.x>.

Gout E, Rébeillé F, Douce R, Bligny R. **Interplay of Mg²⁺, ADP, and ATP in the Cytosol and Mitochondria: Unravelling the Role of Mg²⁺ in Cell Respiration.** *Proceedings of the National Academy of Sciences* 111, 2014: E4560–67. <https://doi.org/10.1073/pnas.1406251111>

Szatmári D, Sárkány P, Kocsis B, Nagy T, Miseta A, Barkó S, Longauer B, Robinson RC, Nyitrai M. **Intracellular Ion Concentrations and Cation-Dependent Remodelling of Bacterial MreB Assemblies.** *Scientific Reports* 10, 2020: 12002. <https://doi.org/10.1038/s41598-020-68960-w>.

a.

b.

Fig. Rev1_1 | Effect of magnesium ion concentration on the GTP/ATP inhibitory effect of *Msm*IMPDH. The IMPDH reaction was continuously measured at 340 nm in reaction buffer (50 mM HEPES (pH 7.5), 200 mM KCl, 5 mM DTT, 100 μM IMP, 2 mM NAD⁺ and 20 nM *Msm*IMPDH) at 37 °C in a 1-cm quartz cuvette for 10 min. **a.** Dependence of the relative velocity on the ATP concentration at fixed 500 μM GTP at various concentrations of Mg²⁺. **b.** Dependence of the relative velocity on the GTP concentration at fixed 250 μM ATP at various concentrations of Mg²⁺. MgCl₂ was used as source of the Mg²⁺ ions. (●) – reaction without magnesium ions; (■) – reaction with equimolar Mg²⁺ concentration in respect to sum of [ATP] and [GTP] (▲) – reaction with 1 mM Mg²⁺ excess with respect to the sum of both [ATP] and [GTP] (▼) – reaction with 2 mM Mg²⁺ excess in respect to the sum of both [ATP] and [GTP]; $n = 2$.

Fig. Rev1_2 | Effect of magnesium ion concentration on the GTP/ATP inhibitory effect of *MsmIMPDH* IMP reaction kinetics. The IMPDH reaction was continuously measured at 340 nm in reaction buffer (50 mM HEPES (pH 7.5), 200 mM KCl, 5 mM DTT, 2 mM NAD⁺ and 10 nM *MsmIMPDH*) without (○, □ and Δ) or with 500 μM ATP and 500 μM GTP (●, ■, ▲ and ▼) at varied concentrations of IMP and Mg²⁺ in a 1-cm quartz cuvette for 10 min at 37 °C. We performed control reactions without Mg²⁺ (○), with 1 mM Mg²⁺ (□), and with 2 mM Mg²⁺ (Δ); *n* = 2.

For the GTP or ppGpp inhibitory studies (Sup Figures 9 and 12), why was an IMP concentration (2 mM) chosen to be 14 times higher than the measured K_{0.5} and an NAD concentration (100 μM) only 0.1 that of the K_m value?

Response: We thank the reviewer for pointing out this issue. We apologize for the confusion, as the values of IMP and NAD concentrations in the legends were unfortunately switched by mistake. The correct values of 100 μM IMP and 2 mM NAD⁺ have been changed in all relevant places in the text (Fig. 5, Supplementary Figs. 12 and 16).

Minor points:

- throughout the manuscript, several spelling, grammatical, and stylistic errors need to be corrected. They are also weird notations for the enzymatic part (V_{lim} or K_h), the conventional notations (V_{max} and K_{0.5}) should be used. Check the notation for the phosphate buffers: what is KPO4? Tris pH8 with HCl (if yes, then Tris-HCl)?

In the “IMPDH phylogenetic analysis”, correct the GuaB (and not Guab or GuB); in this same §, if genes are taken into consideration instead of proteins, then the notation should be guaB (italics and g in lower case). Again, in this §, last sentence, what “j the index of the character in the character’s alphabet” means?

Response: We have fixed all of the above issues. The enzymology notations have been changed to the more precise or common “*k_{cat, app}*” and “*K_{0.5}*”. The buffers and the phylogenetic analysis notations have been corrected accordingly. “*J*” denotes the number of characters in the alphabet

(typically 20 for amino acids). “ p_{wj} ” represents the probability of the j^{th} amino acid (or character in the general sense) appearing at position w in the sequence motif. “ j ” indexes these characters, meaning it iterates over all possible amino acids (or characters) in the alphabet to calculate the sum of their information contributions at position w .

This approach enables a specific position (w) within a protein sequence to be quantified based on the conservation of amino acids across a set of protein sequences. The more conserved the position – in other words, the more frequently a single amino acid occurs in that position across the sequences analyzed – the higher its information content, ranging from zero bits, indicating no conservation (all amino acids equally likely to occur), to a maximum of four bits for absolute conservation (only one amino acid present).

- *Figure 5 should be modified so that all the residues listed in the text are also represented in the different panels. The angle values in panel d should be added.*

Response: The figure has been changed accordingly. The only residue currently not shown in Fig. 5 is R236, as it lies outside the field of view in the respective figure.

- *Figure 6 is confusing. It seems that IMPDH is tetrameric and that in the presence of IMP the tetramer dissociates into dimers.*

Response: The figure has been updated to better reflect the oligomeric state of the enzyme. We thank the reviewer for pointing out this potential source of confusion.

Reviewer #2 (Remarks to the Author):

This manuscript describes mechanisms of allosteric regulation of mycobacterial IMPDH by adenine and guanine nucleotides - a well-studied phenomenon in other species, but here there is a clear description of the mechanism by which ATP/GTP ratios are sensed by the enzyme that is novel and should be of specific interest to the field. It also establishes a role for ppGpp in regulating mycobacterial IMPDH. More broadly, the description of complex structural regulatory mechanisms is likely to be of interest to a wider audience interested in principles of allosteric regulation.

The structural studies by cryo-EM are well done, and support the conclusions. Functional enzyme activity data and mutational analysis are consistent with the structural data, and HDX is used to confirm relative stability of regions that is consistent with the cryo-EM structures.

Response: We thank the reviewer for the thoughtful analysis and commentary.

The paper is well written and clear. I have only a few minor comments that I think may improve the analysis:

1) The findings will be of interest to IMPDH aficionados generally, and are consistent with multiple structural studies of eukaryotic IMPDH in recent years. The discussion would benefit significantly from a comparison of the structural rearrangements observed here with those seen in eukaryotic IMPDHs, specifically:

Response: We fully agree with the reviewer that the comparison of our results with eukaryotic IMPDHs could benefit the manuscript. The corresponding comparative analysis has been added to the Discussion section (lines 456-479) and Supplementary Fig. 17 and 18.

- how do the compressed GTP/ppGpp bound structures compare with eukaryotic compressed structures? Does the compressed or "super-compressed" conformation more closely resemble those?

Response: In general, both the compressed and super-compressed *Msm*IMPDH conformations are remarkably similar, differing only in the degree of compression and the presence of the swapped flap region. While the swap is unique and has never been reported, the actual relative degree of compression differs slightly in all published IMPDHs structures. As a result, there is no discrete comparison for our compressed/super-compressed conformation. For example, human IMPDH2 (PDB code 6uc2) degree of compression is roughly in between the *Msm*IMPDH compressed/super-compressed conformations.

The similarity of the compressed states is now mentioned in the text (lines 471-472).

- variability if the conformational state of ATP/IMP-bound human IMPDH was reported several years ago (Johnson & Kollman, eLife 2020). How do the ATP/IMP-bound mycobacterial structures compare to those in terms of extension/compression of the interdomain hinges?

Response: We thank the reviewer for the thought-provoking question. The Discussion section has been modified to include an additional comparison of the hinge lock mechanisms of eukaryotic and mycobacterial IMPDHs (new Supplementary Fig. 18; lines 471-479). While the binding sites of GTP and ppGpp are different in these different IMPDHs, the general mechanism of how the hinges are locked appears remarkably similar.

Notably, we observed several bent-like conformations in minor classes in our early cryo-EM preliminary datasets with lower ligand concentrations (300 μ M as opposed to 2–10 mM used in the final experiments). The respective resolutions of these classes were too low to be meaningfully interpreted and the aforementioned screening datasets were not included in the final manuscript. It is unclear whether such bent-like conformations play any role in the dynamics of bacterial IMPDHs. It is tempting to speculate that similar half-extended octamers could be an intermediate step in octameric extension, possibly featuring only partially occupied ligand/substrate binding sites.

- the descriptions of ligand binding (IMP/ATP/GTP/ppGpp) are very clear, but it would be useful to know how these compare to prior structures with these ligands bound (one assumes they are identical, but if not this would be very interesting)

Response: The mode of ligand binding is indeed nearly identical with other published bacterial and eukaryotic IMPDH structures. The description in the main text now highlights this similarity (lines 284-286 and 349-351).

- are the rearrangements of finger, flap, and Cys-loops conserved across these structures?

Response: The overall conformation rearrangement of the finger, flap, and Cys loops upon IMP binding remarkably similar to other published bacterial (for example Ref 19 and 20) and eukaryotic structures (for example Ref 3 and 18). The rearrangement also tallies with the canonical pre-reaction substrate-bound state reported in the work by Josephine et al. (Ref 17) This clarification is now included in the main text (lines 284-286).

2) It would be interesting to discuss whether there are conformational changes in the arrangement of the tetrameric catalytic domains that accompany IMP binding and/or ATP/GTP binding. Two conformations were reported in the Johnson & Kollman paper (and subsequent papers) associated

with different activity levels, described as "bowed" and "flat" and I'm wondering if a similar transition occurs in the mycobacterial structures.

Response: We thank the reviewer for the interesting question.

While all of our *Msm*IMPDH structures are not directly comparable to the flat or bowed conformations seen in eukaryotic IMPDHs, they also show some degree of flexibility within the tetramer, reminiscent of the flat and bowed conformational transitions (new Supplementary Fig 17; lines 460-470 in the Discussion section).

Notably, the flexibility described in *Msm*IMPDH is comparably insignificant and does not involve any pronounced structural changes analogous to N-terminal rearrangement of human IMPDHs following the flat or bowed transition. In the case of non-filamenting IMPDHs (similar changes were also noticed by Buey et al., Prot. Science 2022), there does not seem to be any obvious regulatory role for such conformational changes. We can only assume that this flexibility is a macroscopic consequence of the overall conformational changes of the octamer in response to ligand and substrate binding, and possibly during the catalytic cycle. Such conformational plasticity may have been later exploited for a regulatory role in the evolutionarily younger eukaryotic filamenting IMPDHs.

3) In figure 4 it would be helpful to specify for each panel exactly which structures are being compared - I was unsure, for example, in 4a whether these were two different classes from the ATP/IMP dataset, or this was ATP vs ATP/IMP?

Response: We thank the reviewer for pointing out the potentially confusing presentation in Fig. 4. The structures used for the comparison have been newly specified in the figure legend.

4) The authors present new data (Suppl. Fig. 14) in the discussion which is not discussed in results. I understand that the results are somewhat tangential to their overall story, but these should be introduced somewhere in the results section (or omitted - but I think this will be useful information for the field).

Response: We have now included the additional data (previously presented in Supplementary Fig. 14) to the Results section (lines 155-158, now renumbered as Supplementary Fig. 7).

Reviewer #3 (Remarks to the Author):

The manuscript by Bulvas et al describes the allosteric regulation of mycobacterial IMPDH2. Although this enzyme has received attention as a promising target for new antimycobacterial agents, the effects of ATP, GTP and ppGpp on enzymatic activity have not been previously investigated. Thus this work is a very important contribution. The manuscript reports a thorough characterization of allosteric regulation, including enzyme kinetics, H/D exchange and cryoEM, with the unanticipated finding of ATP-dependent GTP inhibition that is further enhanced by ppGpp. The cryoEM and H/D exchange experiments provide the structural context for these findings. The experiments are carefully performed and the paper is very well written. Below are a couple questions/comments that should be addressed.

Response: We thank the reviewer for the concise summary of our work.

Major points

1. The values for K_m , nH etc should be included in the text.

Response: All the basic kinetic parameters have been incorporated into the main text (lines 136-137) in addition to Supplementary Tables 1 and 2.

2. Hill kinetics for IMP dependence have not generally been observed in other IMPDHs. The authors are careful to include constant excess Mg relative to the ATP/GTP/ppGpp concentrations. However, IMP and NAD can also chelate Mg. Could Mg be the cause of the cooperativity? Is cooperativity observed in the absence of Mg?

Response: We thank the reviewer for raising this interesting point. IMP-dependent Hill kinetics have been reported for *M. tuberculosis* IMPDH (Rostirolla et al., 2014) as well as for IMPDHs of *Legionella pneumophila*, *Neisseria meningitidis*, and *Pseudomonas aeruginosa* (Alexandre et al., 2015). However, the exact mechanism underlying the IMP cooperativity is not understood. The concentration of free Mg^{2+} in the cytoplasm of bacteria varies between 1 and 2 mM, depending on the cultivation conditions and genetic background (Alatossava et al., 1985; Froschauer et al., 2004; Szatmári et al., 2020), with nucleic acids and nucleotides the main chelators (Gout et al., 2014). In general, dinucleotides and mononucleotides are much weaker Mg^{2+} binders than triphosphates. The K_d of the $MgNAD^+$ complex (Apps, 1973) is roughly three orders of magnitude greater than $MgATP$ (Gout et al., 2014) (20 mM vs. 30 μ M, respectively). On the other hand, we agree with the reviewer that the effect of Mg^{2+} on the IMP-dependent Hill kinetics of *Msm*IMPDH via its direct effect on the enzyme and/or indirectly via substrate/product chelation cannot be excluded.

To test this possibility, we measured *Msm*IMPDH IMP kinetics in the presence of different fixed Mg^{2+} concentrations (see Fig. Rev3_1 A below) as well as, conversely, relative velocity at fixed IMP concentrations (100 μ M and 500 μ M) across a range of Mg^{2+} concentrations (Fig.

Rev3_1 B). The data show that Mg^{2+} slightly increases the apparent $K_{0.5}$ values and decreases $k_{cat, app}$ at higher concentrations (Fig. Rev3_1 A). This conclusion was partially verified by an independent experiment employing a different design (Fig. Rev3_1 B). However, we observed no significant change in the Hill index up to 5 mM Mg^{2+} . Based on these results, we conclude that Mg^{2+} does not influence IMP-driven *Msm*IMPDPH cooperativity. The change in the $K_{0.5}$ and $k_{cat, app}$ values in the presence of Mg^{2+} ions may be related to their competition with K^+ ions in the regulatory region close to the NAD^+ binding site, as previously reported for *Escherichia coli* IMPDPH (Kerr et al., 2000). Please note that these figures are not included in the current version of the manuscript. Our only intention was to fully address the comment raised by the reviewer.

- Alatossava T, Jütte H, Kuhn A, Kellenberger E. **Manipulation of Intracellular Magnesium Content in Polymyxin B Nonapeptide-Sensitized *Escherichia Coli* by Ionophore A23187.** *Journal of Bacteriology* 162, 1985: 413–19. <https://doi.org/10.1128/jb.162.1.413-419.1985>.
- Alexandre T, Rayna B, Munier-Lehmann H. **Two Classes of Bacterial IMPDPHs According to Their Quaternary Structures and Catalytic Properties.** *PLOS ONE* 10, 2015: e0116578. <https://doi.org/10.1371/journal.pone.0116578>.
- Apps DK. **Complex Formation between Magnesium Ions and Pyridine Nucleotide Coenzymes.** *Biochimica et Biophysica Acta (BBA) - General Subjects* 320, 1973: 379–87. [https://doi.org/10.1016/0304-4165\(73\)90319-X](https://doi.org/10.1016/0304-4165(73)90319-X).
- Froschauer, EM, Kolisek M, Dieterich F, Schweigel M, Schweyen RJ. **Fluorescence Measurements of Free $[Mg^{2+}]$ by Use of Mag-Fura 2 in *Salmonella Enterica*.** *FEMS Microbiology Letters* 237, 2004: 49–55. <https://doi.org/10.1111/j.1574-6968.2004.tb09677.x>.
- Gout E, Rébeillé F, Douce R, Bliigny R. **Interplay of Mg^{2+} , ADP, and ATP in the Cytosol and Mitochondria: Unravelling the Role of Mg^{2+} in Cell Respiration.** *Proceedings of the National Academy of Sciences* 111, 2014: E4560–67. <https://doi.org/10.1073/pnas.1406251111>
- Kerr KM, Cahoon M, Bosco DA, Hedstrom L. **Monovalent Cation Activation in *Escherichia Coli* Inosine 5'-Monophosphate Dehydrogenase.** *Archives of Biochemistry and Biophysics* 375, 2000: 131–37. <https://doi.org/10.1006/abbi.1999.1644>.
- Rostirolla DC, de Assunção TM, Bizarro CV, Basso LA, Santos DS. **Biochemical Characterization of *Mycobacterium Tuberculosis* IMP Dehydrogenase: Kinetic Mechanism, Metal Activation and Evidence of a Cooperative System.** *RSC Advances* 4, 2014: 26271–87. <https://doi.org/10.1039/C4RA02142H>.
- Szatmári D, Sárkány P, Kocsis B, Nagy T, Miseta A, Barkó S, Longauer B, Robinson RC, Nyitrai M. **Intracellular Ion Concentrations and Cation-Dependent Remodelling of Bacterial MreB Assemblies.** *Scientific Reports* 10, 2020: 12002. <https://doi.org/10.1038/s41598-020-68960-w>

$[MgCl_2]$ (mM)	$k_{cat, app}$ (s^{-1})	$K_{0.5}$ (μM)	n_h
0	2.08 ± 0.06	142 ± 6	2.12 ± 0.13
1	2.22 ± 0.04	148 ± 3	1.90 ± 0.06
2	2.24 ± 0.08	151 ± 8	1.91 ± 0.13
5	1.99 ± 0.10	169 ± 12	1.90 ± 0.16
10	1.87 ± 0.05	210 ± 8	1.87 ± 0.06

- 100 μM IMP
- ▲ 500 μM IMP

Fig. Rev3_1 | Effect of magnesium ion concentration on *Msm*IMPDH activity. The IMPDH reaction was continuously measured at 340 nm in reaction buffer (50 mM HEPES (pH 7.5), 200 mM KCl, 5 mM DTT, 2 mM NAD⁺, and 20 nM *Msm*IMPDH) at 37 °C in a 1-cm quartz cuvette for 10 min. **a**, IMP kinetics relative to various Mg²⁺ concentrations. Table to the right shows calculated parameters of fitted Hill equation to the data shown on the left. **b**, Dependence of the relative velocity on the Mg²⁺ concentration at fixed IMP concentrations of 100 μM and 500 μ; *n* = 2.

3. The authors would like to argue that GTP inhibition is physiologically relevant. They previously reported the intracellular concentrations of ATP and GTP were 4 mM and 2 mM, respectively, in Knejzlik et al 2021. Therefore it would be nice if Fig.1b and d included GTP = 2 mM.

Response: We have added the additional data points for 2 mM GTP to Fig. 1b.

4. The inactive open conformation of the Cys loop appears to be incompatible with K⁺ binding. Was there K⁺ in the cryoEM experiments? Please include the buffer for these experiments in the Methods.

Response: The cryo-EM methods section has been amended with the information on the buffer composition used (lines 652-653). The composition of the cryo-EM buffer was the same as for the enzymatic assays (50 mM HEPES, pH 7.4, 200 mM KCl, 5 mM DTT, 2 mM MgCl₂ excess over nucleotide ligands). It has been previously reported (Josephine et al. DOI: 10.1021/bi101590c) that the monovalent cation binding site is formed only after formation of the E-XMP* complex. This is probably why the K⁺ ion did not appear in any of our structures, even with the closed conformation of the Cys-loop.

5. line 434-435: Contrary to the author's claim, obtaining selectivity has not been a challenge for inhibitors that bind to the active site of mycobacterial IMPDHs- in fact, both of the papers cited describe how the cofactor sites of human and bacterial IMPDHs are highly diverged. Selective inhibitors that bind in the cofactor site have been reported- for example PMID: 29746130 and PMID: 26440283. That none of these have entered the clinic is due to the many other challenges of drug discovery- which is reason enough to need new approaches.

Response: We thank the reviewer for this clarification. We have revised the relevant sentence to better reflect the published data regarding the selectivity of IMPDH inhibitors (lines 513-519).

6. It would be nice if the authors included a comparison of their findings with the allosteric regulation of the human enzymes.

Response: We agree with the reviewer that comparing our results with eukaryotic IMPDHs would benefit the manuscript. The corresponding comparative analysis has been added to the Discussion section (lines 456-479) and Supplementary Fig. 17 and 18.

The general mechanism behind IMPDH regulation seems to be conserved from bacteria to eukaryotes, but the molecular details differ. Two interesting points emerge from our comparison:

- Two IMPDH conformations described as “bowed” and “flat” have been documented in the case of filamenting IMPDHs. While all of our *Msm*IMPDH structures are not directly comparable to the flat or bowed conformations seen in eukaryotic IMPDHs, they also show some degree of flexibility within the tetramer, reminiscent of the flat and bowed conformational transitions (Supplementary Fig. 17). However, the relative range of those movements is comparatively limited.
- We compared the hinge lock mechanisms of eukaryotic and mycobacterial IMPDHs (Supplementary Fig. 18). While the binding sites of GTP and ppGpp differ in the case of the IMPDHs, the general mechanism behind the locking of the hinges is remarkably similar.

Minor points

1. The experiments with varied IMP were performed at fixed NAD = 2 mM, which is approximately the value of K_m . Since the fixed NAD concentration was not saturating, the experiments do not really measure “ k_{cat} ”.

Response: We have changed all instances of the inaccurate notation in the text with the more accurate notation “ $k_{cat, app}$ ”.

2. It would be nice if the plots showed the concentration of fixed reagents, for example ATP and NAD in e.g., Fig. 1a (as the authors do in the Supplementary figures).

Response: The concentrations of the fixed substrates were added to all the relevant plots (Figs. 1 and 5).

3. line 213: “a notable increase in deuterium uptake” relative to what?

Response: The sentence has been revised to better convey the idea that the HDX rate is higher relative to other parts of the protein within the apo dataset (now line 246).

4. Fig 3d. What are RFU?

Response: We thank the reviewer for pointing out the discrepancy in the notation of the HDX–MS experiment. The original notation of “ Δ RFU” (relative fractional uptake) in Fig. 3 has been changed to “ Δ HDX” to better align with the text.

5. *Fig. 4e: does the distance line simply show 10Å, or does it mean those two helices have moved 10Å? If the former, it might be better to have the line on the far right of the figure rather than after the arrow and remove the "delta".*

Response: We have revised Fig. 4e to avoid the confusing interpretation.

6. *Line 331: the authors should describe what mutants were characterized*

Response: The description of the mutants has been added to the original sentence (now line 385).

7. *Supplementary Table 2, 2nd GTP entry, MgCl₂ should presumably be "2" not "1".*

Response: We thank the reviewer for pointing out this issue. The value has been corrected in the revised manuscript.

8. *The term "Kh" is used throughout the manuscript but K_{0.5} is used in the equation.*

Response: The notation has been changed to "K_{0.5}" in all instances throughout the manuscript.

REVIEWERS' COMMENTS

Reviewer #1 (Remarks to the Author):

Thank you for submitting the revised version of your manuscript. I appreciate the efforts you have made to address most of the points raised in my initial review. However, there are still a few issues that need further clarification and adjustments.

- IMP-induced expansion: it is crucial to clarify that the IMP-induced expansion requires the presence of ATP to form octamers. In the presence of IMP alone, the protein remains tetrameric. This distinction (in particular, in the video) and the figures should be explicitly stated to avoid any confusion.
- Figure 6: please review the schematic to incorporate the results from the mass photometry analysis. Specifically, the apo form should be depicted as 50% tetrameric and 50% octameric. This will provide a more accurate representation of your findings. And in the presence of IMP alone, the protein is tetrameric, which is not clear in panel b of this figure and in the corresponding legend, which should be modified.
- Mutants in ATP-binding site 1: the differences in the oligomeric states for the R157A (90% tetramers for the apo form or in the presence of ATP) and K201A (80% octameric mutants for the apo form or in the presence of ATP) need to be discussed in more detail. Highlight how these mutations affect the oligomerization compared to the wild-type protein. Discuss also the differences in the quaternary structure for the apo form between the SAXS and the mass photometry data.
- Notation for kinetics: ensure consistency in the notations for the kinetic parameters. Use $K_{0.5IMP}$ (with '0.5' as a subscript and 'IMP' in uppercase as a superscript) and nH (with 'n' in lowercase and 'H' in uppercase as a subscript) throughout the text, including in Equation 3 of the Methods section, and Supplementary Figure 4.
- The authors should include as supplementary figures all the kinetic data that they have incorporated only in their reply letter. These data are important and should also be described in few words in the manuscript.

Reviewer #2 (Remarks to the Author):

The authors have addressed all of the points made in my initial review, and I have no further suggestions or concerns.

Reviewer #3 (Remarks to the Author):

The authors have addressed all of my concerns. I recommend that the manuscript be published without further revision.

Reviewer #1 (Remarks to the Author):

Thank you for submitting the revised version of your manuscript. I appreciate the efforts you have made to address most of the points raised in my initial review. However, there are still a few issues that need further clarification and adjustments.

Response: We thank the reviewer for the careful critical reading and for all the helpful suggestions to improve our manuscript.

- IMP-induced expansion: it is crucial to clarify that the IMP-induced expansion requires the presence of ATP to form octamers. In the presence of IMP alone, the protein remains tetrameric. This distinction (in particular, in the video) and the figures should be explicitly stated to avoid any confusion.

Response: We have revised the video to better represent the need for ATP to stabilize the *Msm*IMPDPH octamers. The part on IMP-induced expansion have been extended (time 1:39-1:48) and the altered caption now reads:

“Octamers are stabilised by CBS domains with bound ATP”

“Catalytic domains in octamer interact via finger and flap loops”

“IMP alone leads to tetramerization”

“ATP stabilizes IMP-induced extended octamers”.

Additionally, we have added the explicit notion of ATP in the Fig. 4, panel e).

- Figure 6: please review the schematic to incorporate the results from the mass photometry analysis. Specifically, the apo form should be depicted as 50% tetrameric and 50% octameric. This will provide a more accurate representation of your findings. And in the presence of IMP alone, the protein is tetrameric, which is not clear in panel b of this figure and in the corresponding legend, which should be modified.

Response: The Fig. 6 has been modified to include the tetrameric/octameric equilibrium of the apo form of *Msm*IMPDPH. The legend was revised and includes the following description for the panels a-d):

“a, The CBS domains are free in the apo form that exists in both tetrameric and octameric conformations. c, The binding of ATP to Site 1 stabilizes the dimers of the CBS domains and thus the octameric conformation. b,d, IMP binding drives the formation of the active tetramers (b) or extended octamers stabilized by ATP (d).”

- Mutants in ATP-binding site 1: the differences in the oligomeric states for the R157A (90% tetramers for the apo form or in the presence of ATP) and K201A (80% octameric mutants for the apo form or in the presence of ATP) need to be discussed in more detail. Highlight how these

mutations affect the oligomerization compared to the wild-type protein. Discuss also the differences in the quaternary structure for the apo form between the SAXS and the mass photometry data.

Response: The differences of ATP-binding site mutants oligomeric state is now discussed in the main text (line 172): “Interestingly, some of the mutations may have affected the affinity of CBS domain pairs in the apo form, as evidenced by changes in the oligomeric state of *Msm*IMPDH (Supplementary Fig. 14b)”

We have added a commentary on SAXS and mass photometry data differences (line 230): “The differences in the tetramer:octamer equilibrium observed in the apo form of *Msm*IMPDH by mass photometry and SAXS measurements are likely caused by the dramatically different protein concentrations used in both methods.”

As of note, similar mixture of tetramers/octamers in aop form of bacterial IMPDH was also observed by Gedeon et al. 2023 (DOI:10.1002/pro.4703).

- Notation for kinetics: ensure consistency in the notations for the kinetic parameters. Use $K_{0.5,IMP}$ (with '0.5' as a subscript and 'IMP' in uppercase as a superscript) and nH (with 'n' in lowercase and 'H' in uppercase as a subscript) throughout the text, including in Equation 3 of the Methods section, and Supplementary Figure 4.

Response: We thank the reviewer for pointing out this issue. We have fixed the notation in all relevant instances throughout the manuscript, including Equation 3 and Supplementary Figure 4. To ensure the readability of the equations, we have opted for notation of “ $K_{0.5,IMP}$ ”.

- The authors should include as supplementary figures all the kinetic data that they have incorporated only in their reply letter. These data are important and should also be described in few words in the manuscript.

Response: The kinetic data on Mg^{2+} effect on *Msm*IMPDH have been added as Supplementary Fig. 5, and are now mentioned in the Results section (line 93).

Added sentence: “Notably, Mg^{2+} ions are required for the full inhibitory effect of these effectors (Supplementary Fig. 5).”